# Interaction Modeling with Multiplex Attention

**Fan-Yun Sun**
Stanford University

**Isaac Kauvar**
Stanford University

**Ruohan Zhang**
Stanford University

**Jiachen Li**
Stanford University

**Mykel Kochenderfer**
Stanford University

**Jiajun Wu**
Stanford University

**Nick Haber**
Stanford University

## Abstract

Modeling multi-agent systems requires understanding how agents interact. Such systems are often difficult to model because they can involve a variety of types of interactions that layer together to drive rich social behavioral dynamics. Here we introduce a method for accurately modeling multi-agent systems. We present Interaction Modeling with Multiplex Attention (IMMA)[1], a forward prediction model that uses a multiplex latent graph to represent multiple independent types of interactions and attention to account for relations of different strengths. We also introduce Progressive Layer Training, a training strategy for this architecture. We show that our approach outperforms state-of-the-art models in trajectory forecasting and relation inference, spanning three multi-agent scenarios: social navigation, cooperative task achievement, and team sports. We further demonstrate that our approach can improve zero-shot generalization and allows us to probe how different interactions impact agent behavior.

## 1 Introduction

Modeling multi-agent systems is important for a wide variety of applications, including self-driving cars, crowd navigation, and human-machine collaboration. The dynamics of multi-agent systems can be challenging to model as they are usually governed by various independent types of interactions. Consider modeling crowd navigation at a social gathering, where agents' behaviors might be governed by at least two types of interactions: target destinations (e.g. a person trying to meet up with a friend) and collision avoidance (e.g. navigating a busy sidewalk). In social scenarios like this, humans often appear to navigate and make predictions based on estimated high-level intentions and relations among the other people [16, 45, 46]. Motivated by this observation, we aim to build a forecasting model for multi-agent systems that infers high-level abstractions in the form of graphs, entirely through the task of predicting agent trajectories.

Leading approaches in modeling multi-agent systems, such as Neural Relational Inference (NRI) [22], Relational Forward Model (RFM) [8], and their extensions [12, 27], use graph neural networks (GNNs) to infer edge types for every pair of entities in the interacting systems. However, this inductive bias does not explicitly handle the multiple layers of interactions present in social multi-agent systems and, as shown empirically in Section 4, has led to at least two shortcomings: reduced performance on long-term predictions and decreased interpretability. Our approach, Interaction Modeling with Multiplex Attention (IMMA), overcomes these issues by using a multiplex graph[2] latent structure to model multiple interaction types, with attention graph layers that can capture the strength of relations.

---

[1] Project website: https://cs.stanford.edu/ sunfanyun/imma/

[2] A multiplex graph is a graph with multiple layers. Nodes are replicated over layers, but each layer can have different connectivity [14].

36th Conference on Neural Information Processing Systems (NeurIPS 2022).

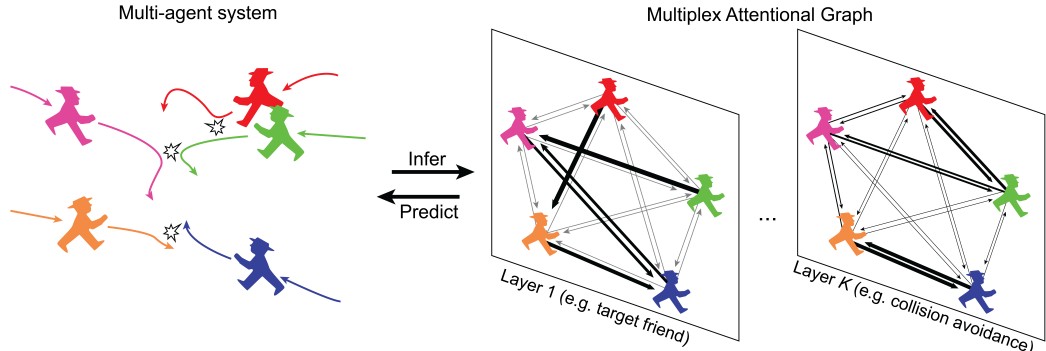

Figure 1: Multi-agent systems are often guided by a variety of interactions, such as target destinations (e.g. a person trying to meet up with a friend) and collision avoidance (e.g. navigating a busy sidewalk). Our model uses a multiplex attentional graph to infer multiple, independent types of relations among agents and yield improved performance in predicting the dynamics of the system. In this figure, strength of interaction is indicated by graph line thickness.

Furthermore, we propose a training approach for IMMA that we call Progressive Layer Training (PLT). Learning in high-dimensional space often leads to highly-entangled representations that are uninterpretable [44], but adding explicit disentanglement objectives often leads to decreased model performance [10, 18]. Inspired by the concept of "starting easy" and curriculum learning [5], we train our model to learn progressively, one latent layer at a time. In addition to improving performance, training IMMA with PLT also improves interpretability—offering the potential to answer questions such as: "Who causes an agent to behave this way?"

Using our method, we conduct experiments in a range of social multi-agent environments inspired by real-world scenarios: social navigation, cooperative task achievement, and team sports. We evaluate models on trajectory prediction, we analyze the relations inferred by our model, and we explore the important role these inferred relations play in making accurate predictions. Additionally, we show that IMMA is effective at zero-shot generalization and conditional generation. To summarize,

- We propose Interaction Modeling with Multiplex Attention (IMMA), a model that uses a new latent space structure which we call Multiplex Attentional Graph. It models relations and utilizes these relations to make forecasting predictions.

- We propose a training technique that is enabled by the multiplexed latent structure: Progressive Layer Training (PLT). PLT enables our model to learn types of interactions one at a time and furthers the performance and interpretability of IMMA.

- Empirically, we show that our method (IMMA w/ PLT) outperforms the state-of-the-art models on three different social multi-agent environments. To understand the interpretability of our approach, we evaluate our method on relational inference and conduct a conditional generation experiment. We also compare various ablated and modified versions of our method, including incorporation of a well-established approach for learning disentangled representations, to understand the contribution of each components. Furthermore, we showcase the superior sample efficiency and generalization ability of our model.

## 2 Related Work

Many approaches have been proposed to model multi-agent interactions, especially as a crucial step toward achieving better performance in tasks such as model-based reinforcement learning [37, 48], trajectory prediction [34], and motion planning [40]. In particular, social dynamics modeling has been studied in domains such as human crowds [34], team sports [12, 22, 27], and traffic participants [35] where the interactions between entities lead to highly complicated motion or behavior patterns. Many earlier works, such as Social Forces [17], are deterministic regression models that model humans as objects affected by forces. The rise of deep learning ushered in various neural network based models such as Social-LSTM [1]—an LSTM model with a social pooling layer, social-GAN [13]—a GAN combined with sequence-to-sequence model, Social-Attention [42]—a recurrent neural network that makes use of attention mechanism, and others [11, 25]. Researchers also proposed to model

interactions with networks that were inspired by a Theory of Mind framework [32, 33]. Our work is different as we explicitly infer an interpretable discrete structure and perform inference over it.

In recent years, graph representation learning has been investigated for relational reasoning in multi-agent systems, where nodes represent interacting entities and edges represent their relations [4]. Most existing graph-based methods infer interactions implicitly [3, 7, 19, 26, 29, 35, 36], meaning that little to no interpretable abstractions can be extracted from the models. Other works take pre-defined graph topology based on heuristics as input [1, 23]. NRI [22], DNRI [12], RFM [39], and EvolveGraph [27] takes a step forward to reason about relations by inferring the topology of the underlying interaction graphs. NRI [22] takes the form of a variational auto-encoder while RFM[39], having near-identical architecture as NRI, simply uses a forward prediction framework. All these methods model interactions as latent interaction graphs and leverage them to make future predictions. Our work falls into this line of research with significant improvements to performance in terms of both relational inference and trajectory prediction because of (1) the novel multiplexed latent structure, (2) the incorporation of an attention mechanism, and (3) a progressive training strategy.

# 3 Methods

In this section, we define our problem and introduce our proposed model (IMMA) and training strategy (PLT). Our input consists of trajectories of $N$ agents. We denote by $\mathbf{x}_i^t$ the feature vector of agent $i$ at time $t$, e.g. location. We denote by $\mathbf{x}^t = \{\mathbf{x}_1^t, ..., \mathbf{x}_N^t\}$ the set of features of all $N$ agents at time $t$ and by $\mathbf{X}^{T_1:T_2} = \{\mathbf{x}_i^{T_1:T_2}, i = 1, ..., N\}$ the set of features covering time steps from $T_1$ to $T_2$. The task is to take trajectory observations as input and learn to predict successive trajectories of all agents. We aim to estimate $p(\mathbf{X}^{T_h+1:T_h+T_f}|\mathbf{X}^{1:T_h})$, where $T_h$ is the historical horizon and $T_f$ is the forecasting horizon. Note that even though agents can have relations and goals that vary in different instances, these must be inferred entirely through the supervision signals of the trajectory prediction task (with no supervision of the relations themselves).

## 3.1 Interaction Modeling with Multiplex Attention

We propose Interaction Modeling with Multiplex Attention (IMMA) - a model that consists of an encoder and a decoder trained jointly. The encoder takes a sequence of observations (i.e. trajectories) as input and outputs a latent interaction graph. Subsequently, the decoder predicts future trajectories by performing message passing among agents (nodes) with the inferred latent interaction graph. IMMA assumes that the dynamics of multi-agent systems can be abstracted to a latent graph structure (i.e. a vector of categorical latent variables) $\mathbf{z} = \{z_{ij}\}$, where $z_{ij}$ represents relations between agents $i$ and $j$. We optimize the variational lower bound, as in the conditional variational autoencoder (CVAE) [38, 21]:

$$\mathcal{L}(\theta, \phi) = \mathbb{E}_{q_\phi(\mathbf{z}|\mathbf{X}^{1:T_h+T_f})}[\log p_\theta(\mathbf{X}^{T_h+1:T_h+T_f}|\mathbf{X}^{1:T_h}, \mathbf{z})] - D_{KL}(q_\phi(\mathbf{z}|\mathbf{X}^{1:T_h+T_f}) \parallel p(\mathbf{z})). \quad (1)$$

The encoder (approximated posterior) $q_\phi(\mathbf{z}|\mathbf{X}^{1:T_h+T_f})$ is a neural network with parameters $\phi$, the decoder (likelihood) $p_\theta(\mathbf{X}^{T_h+1:T_h+T_f}|\mathbf{X}^{1:T_h}, \mathbf{z})$ is a neural network with parameters $\theta$, and $p(\mathbf{z})$ denotes the prior[3].

In previous works [22, 27, 39, 12], the latent graph $\mathbf{z}$ is inferred by performing edge-type classification. That is, $\mathbf{z}$ represents the likelihoods of edges in the latent graph belonging to a set of classes. However, in social dynamical systems multiple types of *independent* interactions can coexist, with different strengths of relations within an interaction type. In this case, representing the interactions with a single graph with many edge types requires many more edge types to accommodate all possible combinations of interactions. More concretely, consider a navigational environment consisting of two independent types of relations: *friendship* (i.e. target-seeking interaction) and *proximity* (i.e. collision-avoidance interaction). To model this environment with a single graph with many edge types, intuitively we might expect to need at least four edge classes to represent the interactions (*none*, only *friendship*, only *proximity*, and *friendship + proximity*) and many more if we want to model the interactions with higher fidelity (e.g. treating *strong friendship* differently from *weak friendship*). Furthermore, since $\mathbf{z}$ is learned to encode likelihood of an edge belonging to a particular class, it may struggle to capture relative strengths of interactions between agents.

---

[3]When the latent space is sufficiently regularized by construction, we could impose no prior on the latent space, omit the KL divergence term in Eq. (1), and simply train a forward prediction model as in [39].

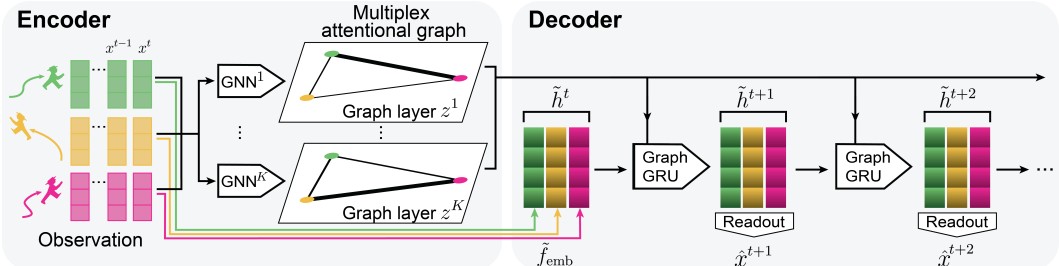

Figure 2: Architecture of IMMA. Each agent's trajectory (e.g. past position across time) is independently embedded into a latent node state. A multiplex graph latent representation is inferred from *all* agents' trajectories. IMMA is trained using Progressive Layer Training, where the GNN to compute layer one is first learned and then a second GNN is introduced to compute layer two (while the GNN weights for layer one is frozen). The decoder is a GraphGRU that uses the same multiplex graph at each prediction step, while updating the latent node state. Note that the input node representations to the decoder are agent-centric. A readout network predicts $\hat{x}$ from $\tilde{h}$ at each time step.

In the following, we describe how IMMA (depicted in Fig. 2) tackles these limitations, as we introduce the encoder and decoder in more detail.

**Encoder with Multiplex Attentional Latent Graphs** The goal of the encoder is to infer a latent interaction graph. To achieve this, we use a GNN on a fully-connected directed graph without self-loops, where each vertex represents an agent. Let $\mathbf{x}_j$ denote the feature vector of agent $j$ across all input frames $(\mathbf{x}_j^1, ..., \mathbf{x}_j^{T_h})$. The GNN embeds agent trajectories (e.g. $\mathbf{x}_j$) into edge embeddings, which are then passed through a MLP to derive unnormalized edge weights (e.g. $\mathbf{e}_{(i,j)}$). $\mathbf{e}_{(i,j)}$ is a vector with dimensionality $K$, a hyperparameter that determines the number of layers in the latent graph. Details of the GNN can be found in the Appendix.

Previous approaches attempt to classify edges into one out of the $K$ possible edge types. They derive $\mathbf{z}_{ij}$ by applying a softmax on $\mathbf{e}_{(i,j)}$ such that $\sum_{k=1}^{k=K} \mathbf{z}_{ij}^k = 1$ with $\mathbf{z}^k$ the $k$-th layer of the latent graph. We propose a novel way to construct the latent space using a multiplex attentional graph. Instead of performing edge-type classification, we perform multi-class prediction on edge embeddings, yielding multiplexed layers, and introduce an attention mechanism between agents within each of these layers. That is,

$$q_\phi(\mathbf{z}_{ij}^k|\mathbf{X}_{1:T_h}) = \text{softmax}_j(\mathbf{e}_{(i,j)}^k) = \frac{\exp(\mathbf{e}_{(i,j)}^k)}{\sum_{j \in N_j} \exp(\mathbf{e}_{(i,j)}^k)}, \qquad \forall\ k = 1\ ...\ K, \tag{2}$$

where $N_j$ denotes the neighbors of agent $j$ (we assume a fully connected graph; thus, $N_j$ includes all other agents except $j$ itself). Note that $\mathbf{z}_{ij}$ is different than $\mathbf{z}_{ji}$. With Eq. 2, layers of the latent graph are *separable* by construction. By separable, we mean that individual layers of the latent graph (e.g. $z^1$ and $z^2$) can be inferred by separate encoders that does not share any parameters. Intuitively, a multiplex latent space allows us to model independent interactions naturally, and the attention mechanism captures strengths of relations.

**Decoder** The task of the decoder is to predict the dynamics of the system using the latent interaction graph $\mathbf{z}$ and the past dynamics. That is, we want to model $p_\theta(\mathbf{X}^{T_h+1:T_h+T_f}|\mathbf{X}^{1:T_h}, \mathbf{z})$. One common issue when training such a decoder is that it might learn to ignore the latent variables while achieving only a marginally worse prediction loss. To avoid this problem, our decoder first learns agent-centric representations $\tilde{\mathbf{h}}_j^t = \tilde{f}_{\text{emb}}(\mathbf{x}_j)$ (i.e. representations are embedded individually for each agent), and then uses these as node embeddings to perform message passing. The decoder thus has to rely on the inferred latent graph to exchange information between agents. After learning agent-centric representations, we use GraphGRU [39] and a readout function to decode future trajectories:

$$\tilde{\mathbf{h}}_j^{t+1} = \text{GraphGRU}(\tilde{\mathbf{h}}_j^t\ \ , \tilde{\mathbf{h}}^t\ \ , \mathbf{z}),\ \hat{\mathbf{x}}_j^{t+1} = \mathbf{x}_j^t\ \ + f_{\text{out}}(\tilde{\mathbf{h}}_j^{t+1}) \tag{3}$$

$$\tilde{\mathbf{h}}_j^{t+2} = \text{GraphGRU}(\tilde{\mathbf{h}}_j^{t+1}, \tilde{\mathbf{h}}^{t+1}, \mathbf{z}),\ \hat{\mathbf{x}}_j^{t+2} = \hat{\mathbf{x}}_j^{t+1} + f_{\text{out}}(\tilde{\mathbf{h}}_j^{t+2}) \tag{4}$$

The details of a GraphGRU block are elaborated in the Appendix. Unlike a typical Graph Attention Network [41] where the graph at every message passing layer is inferred, the same (multiplex) graph

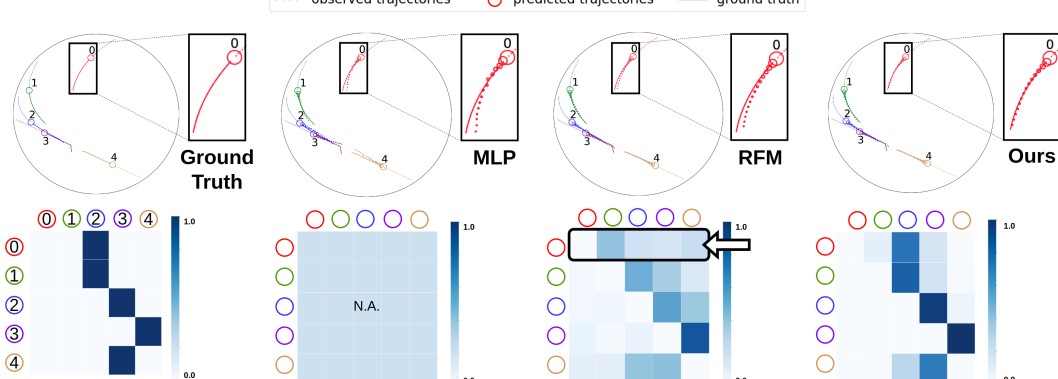

Figure 3: Visualization of the latent graph and agent trajectories of the Social Navigation Environment. (top) Predicted trajectories. The smaller the circle, the further it is into the future. The leftmost column shows ground truth trajectories and the ground truth graph used to simulate those trajectories. (bottom) Inferred latent graphs. Edge strength between agent $i$ and $j$ is represented by darkness of the cell at row $i$ and column $j$. The red agent's relational prediction is inaccurate with RFM—in the row highlighted by the arrow, the green agent is incorrectly given higher weight than the blue agent—and thus the predicted trajectories deviate from the ground truth, especially on long-horizon predictions.

is used across all decoding steps. We can re-evaluate the encoder for every time step to obtain a dynamically changing latent graph, however we leave this to future work.

## 3.2 Progressive Layer Training

The role of the encoder is to infer a set of graph bases $(\mathbf{z}^1, \mathbf{z}^2 ...)$ that reflect the underlying dynamics of the environment. Since the messages are first passed within layers before summing up at the end, it would be useful for the latent graph layers to represent different types of interactions between agents.

However, it is difficult for the model to learn all the interactions, let alone have them be disentangled. Disentanglement is typically achieved by imposing additional loss, yet these approaches tend to come with the trade-off of decreased reconstruction quality [10, 18, 24]. Motivated by curriculum learning [5] and progressive learning [28] (or more broadly the concept of "starting easy"), we propose to overcome this trade-off with Progressive Layer Training (PLT) — a strategy of learning a set of good graph bases by learning the high-level and most consequential interactions first and then progressively growing the network to model lower-level and more intricate interactions. We first train a IMMA with a single-layered latent graph $\mathbf{z}^1$, then add a second layer $\mathbf{z}^2$ by training a second encoder while freezing the weights of the first. This process can be repeated until no further improvement is observed. The fact that we can choose to have a separable latent space and do not have to share any parameters between these encoders is a result of our latent space being multiplexed. Directly adding new components to a trained network can introduce a sudden shock to the gradient. To stabilize training, we adopt "fade-in" [20] to smoothly blend the new and existing network components. Refer to the Appendix for more details.

## 4 Experiments

We design our experiments to answer the following questions:

**Q1**: Does our approach (IMMA w/ PLT) consistently outperform prior methods across a diverse set of social multi-agent environments and benchmarks?

**Q2**: Does the use of multiplex attentional latent graph give more interpretable abstractions?

**Q3**: How much does each component contribute to the final performance?

**Q4**: Is our approach sample efficient? Can it generalize well to novel environments?

To answer **Q1**, we test our approach in three multi-agent environments that each exhibited multiple types of social interaction: our simulated Social Navigation Environment, PHASE [30], and the NBA dataset (used in [22, 27, 49, 47, 15]). We develop the first benchmark ourselves, while the latter two are established benchmarks. For each environment, the model receives multi-agent trajectories of length 24 as observation and predicts 10 future time steps. For Social Navigation we use 100k multi-agent trajectories (in total for training, validation, and testing), for NBA dataset we use 300k multi-agent trajectories, and for PHASE we use 836 multi-agent trajectories. We also perform data augmentation, as described in the Appendix. To answer **Q2**, we evaluate the inferred interaction graphs for the Social Navigation Environment and PHASE, since we have the ground truth interaction graph. Note that we are *not* supervising on any explicit relations of agents. We further conduct qualitative experiments to understand the abstractions provided by the latent graph. To answer **Q3**, we conduct an ablation study. For **Q4**, we perform a sample efficiency experiment and an experiment of zero-shot generalization. Experiments for **Q3** and **Q4** are conducted on the Social Navigation Environment as we have full control over it.

## 4.1 Experiment Setup

Here we describe the environments and baselines. More details are in the Appendix.

**Social Navigation Environment**   The Social Navigation Environment simulates a socially inter-active crowd where agents seek to meet up with other agents while simultaneously navigating to avoid collisions within the crowd (as schematized in Fig. 1). This environment allows fine-grained control by simulating lower-level interaction dynamics using Optimal Reciprocal Collision Avoidance (ORCA, [6]) like many other crowd navigation works [8, 9]. With this environment, we can easily vary environmental parameters to generate diverse trajectories for our generalization experiment. We experiment primarily with an environment consisting of five agents. For each simulated episode, a social interaction graph of the agents is randomly generated: each agent is assigned to be the follower of another agent. Agents are initialized to positions along the edge of a circle. Dynamics is simulated using ORCA by assigning the position of each agent's leader (at each time step) as the target destination of that agent. These environment settings yield reasonably complex trajectories that were clearly influenced by the social interaction graph and collision avoidance, as seen in example trajectories of Fig. 3.

**PHASE dataset**   PHASE [30] is a simulator for physically-grounded abstract social events, in which social recognition and social prediction algorithms can be benchmarked. In PHASE, two animated agents move in a continuous 2D space and interact with each other and other objects (2 balls). Their actions are generated using a physics engine and a hierarchical planner [30]. We choose the "collaboration" task for our experiment, in which two agents collaborate to move one of the balls to a pre-specified landmark location. Diverse trials are generated by randomizing the agents' and balls' starting locations as well as orientations, target ball, and target landmark. Note that this dataset is substantially smaller than the two other datasets.

**NBA dataset**   The National Basketball Association (NBA) dataset consists of position trajectories for ten interacting basketball players and the ball and uses real-world player tracking data. The task

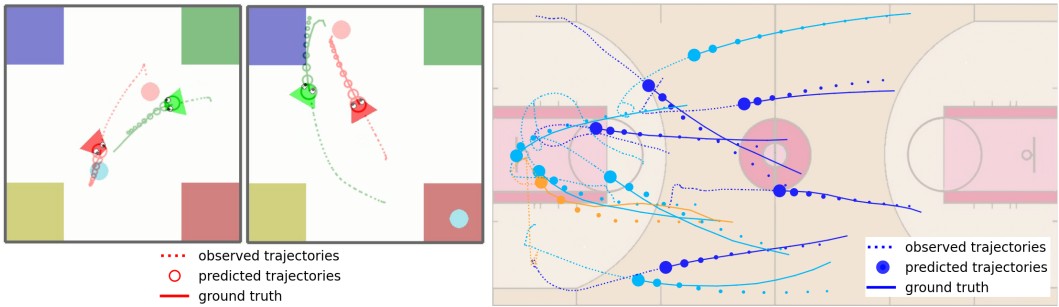

Figure 4: Qualitative analysis of our approach on PHASE data (left) and the NBA dataset (right). For the NBA visualization, the orange agent is the basketball, and colors represent teams.

Table 1: Trajectory prediction results (mean $\pm$ std, over 3 runs) on all three datasets.

| Metrics | Baseline Methods | | | | | | | IMMA (ours) | | |
|---|---|---|---|---|---|---|---|---|---|---|
| | MLP | GAT_LSTM [41] | NRI [22] | EvolveGraph [27] | RFM (skip 1) | RFM [39] | fNRI [43] | SG | MG | MG + PLT |
| Social Navigation environment | | | | | | | | | | |
| ADE ↓ | $0.241 \pm 0.006$ | $0.306 \pm 0.008$ | $0.217 \pm 0.005$ | $0.160 \pm 0.003$ | $0.160 \pm 0.002$ | $0.156 \pm 0.003$ | $0.151 \pm 0.001$ | $0.164 \pm 0.002$ | $0.148 \pm 0.004$ | $\mathbf{0.139 \pm 0.001}$ |
| FDE ↓ | $0.513 \pm 0.012$ | $0.527 \pm 0.009$ | $0.386 \pm 0.009$ | $0.321 \pm 0.006$ | $0.325 \pm 0.004$ | $0.317 \pm 0.005$ | $0.308 \pm 0.003$ | $0.327 \pm 0.003$ | $0.294 \pm 0.006$ | $\mathbf{0.279 \pm 0.002}$ |
| PHASE | | | | | | | | | | |
| ADE ↓ | $1.024 \pm 0.005$ | $1.545 \pm 0.040$ | $0.986 \pm 0.011$ | $0.848 \pm 0.012$ | $0.870 \pm 0.008$ | $0.892 \pm 0.008$ | $0.883 \pm 0.015$ | $0.875 \pm 0.020$ | $0.883 \pm 0.024$ | $\mathbf{0.801 \pm 0.009}$ |
| FDE ↓ | $1.763 \pm 0.024$ | $2.527 \pm 0.063$ | $1.772 \pm 0.024$ | $1.522 \pm 0.019$ | $1.581 \pm 0.030$ | $1.630 \pm 0.042$ | $1.607 \pm 0.044$ | $1.585 \pm 0.044$ | $1.593 \pm 0.048$ | $\mathbf{1.484 \pm 0.014}$ |
| NBA dataset | | | | | | | | | | |
| ADE ↓ | $1.113 \pm 0.002$ | $0.978 \pm 0.005$ | $0.946 \pm 0.005$ | $0.896 \pm 0.009$ | $0.938 \pm 0.003$ | $0.839 \pm 0.020$ | $0.804 \pm 0.004$ | $0.860 \pm 0.008$ | $0.833 \pm 0.017$ | $\mathbf{0.769 \pm 0.009}$ |
| FDE ↓ | $1.990 \pm 0.002$ | $1.733 \pm 0.010$ | $1.818 \pm 0.017$ | $1.695 \pm 0.016$ | $1.756 \pm 0.005$ | $1.572 \pm 0.034$ | $1.517 \pm 0.012$ | $1.612 \pm 0.015$ | $1.562 \pm 0.027$ | $\mathbf{1.438 \pm 0.019}$ |

Table 2: Relational inference accuracy on the Social Navigation environment and PHASE.

| Metrics | Baseline Methods | | | | | | | IMMA | | |
|---|---|---|---|---|---|---|---|---|---|---|
| | MLP | GAT_LSTM [41] | NRI [22] | EvolveGraph [27] | RFM (skip 1) | RFM [39] | fNRI [43] | SG | MG | MG + PLT |
| (Social Navigation) Graph Acc. (%) ↑ | - | $21.83 \pm 0.67$ | $57.18 \pm 0.14$ | $70.23 \pm 0.36$ | $70.05 \pm 0.55$ | $71.53 \pm 0.56$ | $33.97 \pm 4.89$ | $\mathbf{80.15 \pm 1.48}$ | $77.98 \pm 0.16$ | $\mathbf{81.38 \pm 0.29}$ |
| (PHASE) Graph Acc. (%) ↑ | - | $52.94 \pm 1.66$ | $55.30 \pm 2.09$ | $58.96 \pm 1.31$ | $55.30 \pm 2.09$ | $54.71 \pm 0.83$ | $55.49 \pm 6.03$ | $78.82 \pm 3.54$ | $\mathbf{79.02 \pm 4.04}$ | $79.21 \pm 1.42$ |

is to predict 10 future time steps (4s) based on a history of 24 time steps (10s). Team information is an additional input feature. The unit reported is meters.

**Baselines**   We consider the following baselines and ablations.

- **MLP**: a large MLP network that takes the concatenated features of all agents across all timesteps as input and output trajectory predictions of all agents at the same time.
- **GAT_LSTM**: a network with Graph Attention layers [41] followed by LSTMs.
- **NRI** [22]: a VAE model with recurrent GNN modules.
- **EvolveGraph** [19]:a model that expands upon the framework of NRI and introduces a double-stage training pipeline to account for an evolving interaction graph.
- **RFM** [39]: a recurrent GNN model of similar architecture to NRI, but with a supervised loss instead of a variational lower bound loss.
- **RFM (skip 1)**: a RFM model with the first edge type is "hard-coded" as "non-edge" (i.e. no messages are passed along edges of the first layer).
- **fNRI** [43]: a NRI model that uses a multiplex latent graph structure (i.e., sigmoid activation).
- **IMMA (SG)**: IMMA with a single layer of attentional latent graph.
- **IMMA (MG)**: IMMA with multiple layers of attentional latent graphs (a multiplex graph) trained simultaneously, without PLT.

We are aware that there are other relevant baselines such as Social-LSTM [1], Social-GAN [13], and Trajectron++ [35] etc. We choose to compare with EvolveGraph, because it consistently outperforms these models across datasets [27].

## 4.2   Trajectory Prediction

To answer **Q1**, we evaluate all models on their prediction performance of agent position on future trajectories with two metrics: average displacement error (ADE) and final displacement error (FDE). Both of these are metrics widely used in trajectory prediction literature: ADE is defined as the average deviated distance of all entities within the prediction horizon, and FDE is defined as the deviated distance of all entities at the last predicted time step.

The results on the Social Navigation Environment, PHASE, and the NBA dataset are shown in table 1. We found that using multiplex attentional latent graph with progressive layer training (MG w/ PLT) yields the best performance on all three datasets, outperforming previous state-of-the-art models. On the Social Navigation Environment and the NBA dataset, IMMA without progressive layer training (MG) still performs better than all baselines, showing the effectiveness of using multiplex attentional graph as latent structure. We visualize the prediction results of our method and the best performing baseline (RFM) in Fig. 3 on the Social Navigation Environment. Visualizations of PHASE and the NBA dataset are shown in Fig. 4 (more can be found in the Appendix).

### 4.3 Relational Analysis and Interpretability

To answer **Q2**, we conduct experiments to understand what abstractions can be extracted from the inferred multiplex attentional graph. We first compare the performance of our approach to the same set of baselines used in the previous experiment on relational inference. Intuitively, this is to evaluate how well models can answer the question "What causes an agent to behave this way?" Refer to the Appendix for details on how we calculate the graph accuracy. We conduct this experiment in the Social Navigation Environment and PHASE, since we have access to the ground-truth social interaction graph for these two environments.

IMMA more accurately estimates the underlying social interaction graph (Table 2). The results can also explain the benefits of training IMMA with PLT: by leveraging the graph accuracy prediction that arises from using a single graph layer, but with the expressive power of a full multiplex graph.

To further understand how IMMA uses the inferred social graph, we assess how the trajectory predictions are altered if we manually manipulate the latent graph. More formally, we ask our decoder to generate new trajectories conditioned on the new latent graph $p_\theta(\mathbf{X}^{T_h+1:T_h+T_f}|\mathbf{X}^{1:T_h}, \mathbf{z}^{\text{new}})$. In Fig. 5, we see that with IMMA, changing the leader of an agent clearly alters the predicted trajectory to target that new leader while keeping the predictions for other agents intact, whereas the generated trajectories of the baseline consist of unrealistic turns (red agent 0) and the predicted trajectories of other agents deteriorates at the same time. Using the trajectories shown in Fig. 5, we ran a human study and our model's prediction is judged as more reasonable than the RFM model's in 76.67% of the trials. Find more details of the human study in Appendix section C.

### 4.4 Ablation Study

To answer **Q3**, we experiment with three ablated versions of our model. We ensure that all ablated versions have around the same amount of total parameters and have two layers of latent graphs. Additionally, we measure the information dependency between the two layers of latent graph with Normalized Mutual Information (NMI) score (details in the Appendix). Results are shown in Table 3. The first column in the table shows the performance of a model where the latent graph structure are edge types between all pairs of agents. It has an NMI score of 1.0 (complete dependency) because the second layer of latent graph is simply one minus the first layer of latent graph. We observe that the biggest gain arose from using a multiplex attention graph and Progressive Layer Training furthers the performance. We also see that both components contribute to decreased information dependency

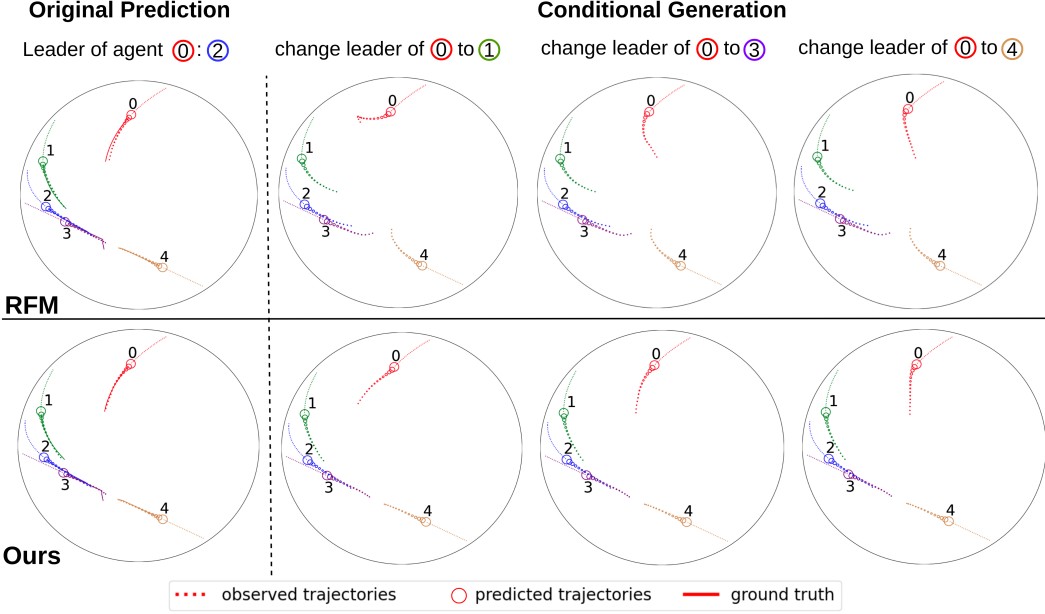

Figure 5: Conditional generation results of the RFM model and our method (IMMA w/ PLT).

Table 3: Ablation study results.

| | Ablated versions of IMMA | | | Ours (IMMA w/ PLT) |
|---|---|---|---|---|
| Multiplex Attention Graph | ✗ | ✔ | ✔ | ✔ |
| Disentanglement Loss [10] | ✗ | ✗ | ✔ | ✗ |
| Progressive Layered Training | ✗ | ✗ | ✗ | ✔ |
| ADE ↓ | $0.156 \pm 0.003$ | $0.148 \pm 0.004$ | $0.197 \pm 0.003$ | $\mathbf{0.139 \pm 0.001}$ |
| FDE ↓ | $0.317 \pm 0.005$ | $0.294 \pm 0.006$ | $0.386 \pm 0.006$ | $\mathbf{0.279 \pm 0.002}$ |
| Graph Acc. (%) ↑ | $71.53 \pm 0.56$ | $77.98 \pm 0.16$ | $69.47 \pm 0.72$ | $\mathbf{81.38 \pm 0.29}$ |
| NMI score ↓ | 1.0 | 0.46 | **0.042** | 0.13 |

Table 4: Zero-shot generalization results on the Social Navigation Environment.

| | Baseline Methods | | | | | | | IMMA (Ours) | | |
|---|---|---|---|---|---|---|---|---|---|---|
| Metrics | MLP | GAT_LSTM [41] | NRI [22] | EvolveGraph [27] | RFM (skip 1) | RFM [39] | fNRI [43] | SG | MG | MG + PLT |
| | 2x speed | | | | | | | | | |
| ADE | $0.303 \pm 0.006$ | $0.361 \pm 0.009$ | $0.269 \pm 0.005$ | $0.219 \pm 0.003$ | $0.209 \pm 0.002$ | $0.205 \pm 0.001$ | $0.206 \pm 0.003$ | $0.213 \pm 0.002$ | $0.197 \pm 0.003$ | $\mathbf{0.192 \pm 0.001}$ |
| FDE | $0.632 \pm 0.011$ | $0.635 \pm 0.011$ | $0.485 \pm 0.009$ | $0.421 \pm 0.006$ | $0.418 \pm 0.005$ | $0.411 \pm 0.005$ | $0.410 \pm 0.001$ | $0.419 \pm 0.004$ | $\mathbf{0.389 \pm 0.006}$ | $\mathbf{0.383 \pm 0.002}$ |
| Graph Acc (%) | - | $22.04 \pm 0.61$ | $55.10 \pm 0.11$ | $67.30 \pm 0.29$ | $68.03 \pm 0.43$ | $69.54 \pm 0.41$ | $33.48 \pm 4.55$ | $\mathbf{77.95 \pm 0.99}$ | $76.07 \pm 0.20$ | $\mathbf{78.84 \pm 0.08}$ |
| | 2x smaller environment | | | | | | | | | |
| ADE | $0.240 \pm 0.006$ | $0.305 \pm 0.010$ | $0.217 \pm 0.005$ | $0.153 \pm 0.003$ | $0.155 \pm 0.001$ | $0.150 \pm 0.003$ | $0.151 \pm 0.003$ | $0.158 \pm 0.002$ | $\mathbf{0.139 \pm 0.003}$ | $\mathbf{0.139 \pm 0.001}$ |
| FDE | $0.509 \pm 0.010$ | $0.526 \pm 0.012$ | $0.386 \pm 0.009$ | $0.314 \pm 0.005$ | $0.312 \pm 0.003$ | $0.303 \pm 0.005$ | $0.308 \pm 0.003$ | $0.312 \pm 0.004$ | $\mathbf{0.275 \pm 0.006}$ | $\mathbf{0.279 \pm 0.002}$ |
| Graph Acc (%) | - | $21.94 \pm 0.62$ | $57.18 \pm 0.14$ | $72.08 \pm 0.39$ | $69.99 \pm 0.49$ | $71.66 \pm 0.64$ | $33.97 \pm 4.89$ | $80.60 \pm 1.44$ | $78.35 \pm 0.05$ | $\mathbf{81.38 \pm 0.29}$ |
| | 2x more agents | | | | | | | | | |
| ADE | - | $1.486 \pm 0.203$ | $0.527 \pm 0.030$ | $0.354 \pm 0.021$ | $0.441 \pm 0.016$ | $0.333 \pm 0.013$ | $0.310 \pm 0.013$ | $0.211 \pm 0.001$ | $0.205 \pm 0.001$ | $\mathbf{0.195 \pm 0.001}$ |
| FDE | - | $2.538 \pm 0.313$ | $1.096 \pm 0.073$ | $0.988 \pm 0.038$ | $0.792 \pm 0.034$ | $0.762 \pm 0.038$ | $0.659 \pm 0.007$ | $0.435 \pm 0.002$ | $0.424 \pm 0.002$ | $\mathbf{0.406 \pm 0.002}$ |
| Graph Acc (%) | - | $19.84 \pm 0.30$ | $34.57 \pm 0.09$ | $50.56 \pm 0.16$ | $49.41 \pm 0.40$ | $51.37 \pm 0.61$ | $19.71 \pm 3.06$ | $62.05 \pm 1.30$ | $62.45 \pm 0.53$ | $\mathbf{64.25 \pm 0.34}$ |

between two latent graph layers without explicitly encouraging disentanglement. Although imposing an additional loss to disentangle between two latent graph layers does yield the smallest NMI score, it comes with the trade-off of decreased forecasting accuracy as observed in other works [10, 18, 24].

## 4.5 Sample Efficiency and Generalization

To answer **Q4**, we investigate sample efficiency and observe that our method consistently outperforms the strongest baseline (RFM) on smaller sample complexity regimes (Fig. 6). We test zero-shot generalization performance in response to three environment modifications: agents moving twice as fast (2x speed), crowding the agents into half as much space (2x smaller environment), and doubling the number of agents (2x more agents). As shown in Table 4, MG and MG+PLT versions of IMMA outperform the baselines in all metrics. Additionally, as shown in Appendix Table 5, the change in performance on the generalization scenarios relative to the original environment was similar to or better than the baselines in all metrics

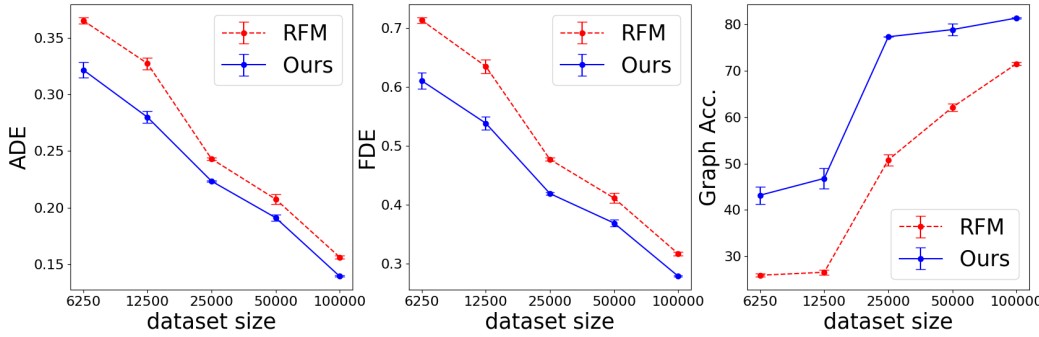

Figure 6: Sample efficiency of RFM and Ours. Lower ADEs and FDEs and higher graph accuracies are better.

## 5 Conclusion

Multi-agent dynamics can often be complex as a result of many layers of underlying social interactions. Agents can be influenced by multiple independent types of relationships with each agent, leading to properties (such as intentionality or cooperation) that are often absent in simpler physical systems. We developed a method that uses multiplex attentional graphs as latent representations to model the dynamics that can arise from such multi-layered multi-agent interactions. Addressing key aspects of social interaction, our method consistently outperforms other state-of-the-art methods in modeling the dynamics of social multi-agent systems. We foresee minimal direct negative societal impact — please see Appendix for further discussion. Future work will extend this approach to active learning settings and investigate using our method to guide model-based reinforcement learning agents.

## 6 Acknowledgements

I.K. is a Merck Awardee of the Life Sciences Research Foundation, and a Wu Tsai Stanford Neurosciences Institute Interdisciplinary Scholar. This work is also in part funded by the Toyota Research Institute (TRI), the Stanford Institute for Human-Centered AI (HAI), ONR MURI N00014-22-1-2740, Meta, Salesforce, and Samsung. N.H. is further funded by the GSE Transforming Learning Accelerator.

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
