## A  Societal Impact

This work has the potential for wide-ranging applications in human-in-the-loop (e.g. assistive and social robotics, self-driving vehicles) and completely autonomous systems. Such systems have the potential for negative societal impact (e.g. harmful dual use), and as researchers we must critically evaluate such applications and promote beneficial ones.

## B  Additional details of our method

**Conditional VAE Formulation**   Note that in the formulation (1), the approximated posterior is conditioned on *all* frames. However, if we train the encoder on all frames, we would need to sample latent variables from the prior at test time. As we would not be able to infer the most likely latent variables based on the historical trajectories (i.e. could not infer relations) [2], that would be undesirable. Thus, we use the form $q_\phi(\mathbf{z}|\mathbf{X}^{1:T_h})$ in place of $q_\phi(\mathbf{z}|\mathbf{X}^{1:T_h+T_f})$.

**GNN**   The GNNs used in the encoder consist of the following message passing operations:

$$\mathbf{h}_j^{(1)} = f_{\text{emb}}(\mathbf{x}_j) \tag{5}$$

$$v{\rightarrow}e: \quad \mathbf{e}_{(i,j)}^{(1)} = f_e^{(1)}([\mathbf{h}_i^{(1)}, \mathbf{h}_j^{(1)}]) \tag{6}$$

$$e{\rightarrow}v: \quad \mathbf{h}_j^{(2)} = f_v^{(1)}(\textstyle\sum_{i\neq j} \mathbf{e}_{(i,j)}^{(1)}) \tag{7}$$

$$v{\rightarrow}e: \quad \mathbf{h}_{(i,j)}^{(2)} = f_e^{(2)}([\mathbf{h}_i^{(2)}, \mathbf{h}_j^{(2)}]) \tag{8}$$

$$... \tag{9}$$

where $\mathbf{x}_j$ is the input trajectory of agent $j$ and $f_{(...)}$ are MLPs. In our experiments we only perform two rounds as message passing even though the above message passing operations can be performed for more than two rounds. $\mathbf{h}_i^{(m)}$ and $\mathbf{e}_{(i,j)}^{(m)}$ denote the embedding of agent $i$ and embeddings of edge $(i,j)$ after $m$ rounds of message passing, respectively. Note that this is not to be confused with $\mathbf{e}_{(i,j)}^k$, which is used to denote the $k$-th element of the vector $\mathbf{e}_{(i,j)}^{(2)}$ in section 3. In a single pass, Eqs. (6)–(8), the resulting edge embedding $\mathbf{e}_{(i,j)}^{(1)}$ only depends on $\mathbf{x}_i$ and $\mathbf{x}_j$, ignoring interactions with other nodes, while $\mathbf{e}_{(i,j)}^{(2)}$ uses information from the whole graph.

**Multiplex Attentional Graph**   In previous works [22, 27, 39, 12], the latent graph (posterior) is inferred by performing edge type classification on the edge embeddings. That is,

$$q_\phi(\mathbf{z}_{ij}^k|\mathbf{X}_{1:T_h}) = \text{softmax}_k(\mathbf{e}_{(i,j)}^k) = \frac{\exp(\mathbf{e}_{(i,j)}^k)}{\sum_{k=1}^{k=K} \exp(\mathbf{e}_{(i,j)}^k)} \tag{10}$$

where $\mathbf{z}^1$ denotes the first layer-graph and $\mathbf{z}_{ij}^1$ denotes the relational strength of agents $i$ and $j$ in the first layer-graph. $\sum_a \mathbf{z}_{ij}^a = 1$ directly follows from the above equation for all $(i,j)$ pairs. We propose a novel way to construct the latent space through multiplex attentional graphs. Instead of performing edge-type classification on edge embeddings, we learn layers of disentangled attentional graph from edge embeddings. Contrast Eq. (2), the latent space construction of our model, to Eq. (10), latent space construction of previous works.

**GraphGRU**   In this paper, we will use $\mathbf{h}$ to represent embeddings in the encoder and $\tilde{\mathbf{h}}$ for embeddings in the decoder. Recall that $\tilde{\mathbf{h}}_j^t = \tilde{f}_{\text{emb}}(\mathbf{x}_j)$ is the agent-centric representations inferred by a MLP $\tilde{f}_{\text{emb}}$. It is used as the input to the GraphGRU in the decoder and $K$ is the number of latent graph layers. The GraphGRU used in our decoder consists of the following operation:

$$\text{MSG}_j^t = \textstyle\sum_{k=1}^{k=K} \sum_{i \in N_j} \mathbf{z}_{ij}^k \tilde{f}_e^k([\tilde{\mathbf{h}}_i^t, \tilde{\mathbf{h}}_j^t]) \tag{11}$$

$$\tilde{\mathbf{h}}_j^{t+1} = \text{GRU}(\text{MSG}_j^t, \tilde{\mathbf{h}}_j^t) \tag{12}$$

$$\hat{\mathbf{x}}_j^{t+1} = \mathbf{x}_j^t + f_{\text{out}}(\tilde{\mathbf{h}}_j^{t+1}) \tag{13}$$

The input to the message passing operation is the recurrent hidden state at the previous time step. $f_{\text{out}}$ denotes an output transformation. For each node/agent $j$ the input to the GRU update is the concatenation of the aggregated messages $\text{MSG}_j^{t+1}$, the current input $\mathbf{x}_j^{t+1}$, and the previous hidden state $\tilde{\mathbf{h}}_j^t$.

**Progressive Layer Training**    we propose Progressive Layer Training (PLT) - a strategy of learning interactions (i.e. latent graphs) of the highest significance to the lowest in a progressive manner. More specifically, we start by only allowing the network to use one layer of latent graph to conduct message passing, and then progressively release more latent graph layers while fixing the previously learned latent layers and their corresponding network components. To do this, we introduce a coefficient $\alpha$ to Eq. (11) when growing new components:

$$\text{MSG}_j^t = \sum_{k=1}^{k=K} \sum_{i \in N_j} \mathbf{z}_{ij}^a \tilde{f}_e^a([\tilde{\mathbf{h}}_i^t, \tilde{\mathbf{h}}_j^t]) + \alpha \sum_{i \in N_j} \mathbf{z}_{ij}^{\text{new}} \tilde{f}_e^{\text{new}}([\tilde{\mathbf{h}}_i^t, \tilde{\mathbf{h}}_j^t]) \tag{14}$$

where $z^{\text{new}}$ is the newly introduced latent graph layer and all the other latent graph layers $z^a$ and their corresponding encoders have their weights fixed. A new encoder $q_{\phi^{\text{new}}}(\mathbf{X}_{1:T_h})$ with parameters $\phi^{\text{new}}$ is introduced to infer $\mathbf{z}^{\text{new}}$. We increase $\alpha$ linearly from 0 to 1 within a certain number of iterations (500 epochs in our experiments).

## C    Human Experiment on the Realism of Figure 5

Using the trajectories shown in Fig. 5, we ran a two-alternative forced (2AFC) choice human study with 20 subjects, in which subjects were asked to choose which model's prediction is more reasonable (RFM vs. ours). In each of the three trials, the order of the predictions is randomized and the subjects do not have access to which prediction is generated by which model. The results show that our model's prediction is judged as more reasonable than the RFM model's in 76.67% of the trials.

## D    Additional Experimental Details and Results

### D.1    Experiment Details

For the Social Navigation Environment, we follow the environmental configurations set in [8, 9][4]. In this simulation, agents are controlled by ORCA [6]. We set the radius of agents to 0.3, the radius of the circular environment to 8, the safety space to 0.09, calculation time horizon (for ORCA) to 1, preferred speed of agents to 1. For Social Navigation we use 100k multi-agent trajectories, in total for training, validation, and testing. We simulate 100k samples in total for training, validation, and testing. For PHASE[5], we choose the "collaboration" task, use environment layout 0 (no walls), and follow the default simulation configuration in [30]. Since the official dataset of PHASE is small, we resort to generating our own dataset with their simulator. 836 multi-agent trajectories are generated by randomizing the agents' and balls' starting locations as well as orientations, target ball, and target landmark, resulting in a diverse set of behaviors. The dataset will be made public. When training models on PHASE, we do data augmentation on the training set by flipping the environment and rotating it by 90%, 180%, and 270%, resulting in a training set with 8x more instances. For the NBA dataset, we use 300k multi-agent trajectories in total[6] for training, validation, and testing. For all dataset, we use 80% for training, 10% for validation, and 10% for testing.

For all experiments and all models, we use a batch size of 64 and use the same convergence criteria: terminate when the performance of the model on the validation dataset has not improved for more than 100 epochs. We use Adam optimizer with an initial learning rate of 1e-6 and decays the learning rate by a factor of 0.9 if the validation performance has not improved in 5 epochs. For the Social Navigation Environment and PHASE experiment, we use all (graph-based) models with 2 latent graph layers and for the NBA dataset we use 5 for all graph-based models (NRI [22],RFM [39],EvolveGraph [27], and IMMA). We use a hidden size of 96, 156, 256 for the Social Navigation Environment, PHASE, and the NBA dataset, respectively. We implement all models and simulations of all environments in PyTorch [31]. All graph-based models use MLP encoders and RNN decoders, both of which

---

[4]https://github.com/vita-epfl/CrowdNav
[5]https://www.tshu.io/PHASE/
[6]https://github.com/linouk23/NBA-Player-Movements

are elaborately described in the Appendix of [22] (section C). The only difference of our model's architecture to theirs is that we use agent-centric representations $\tilde{\mathbf{h}}_j^t = \tilde{f}_{\text{emb}}(\mathbf{x}_j)$ before decoding future trajectories with GraphGRU. We implement $\tilde{f}_{\text{emb}}$ as a 3-layer MLP with batch normalization, dropout, and ELU activations. All trainings are done on a workstation with 8 Nvidia A40 GPUs and 1008G of RAM. We only referenced Github repositories with none or MIT license.

To calculate relational inference accuracy (graph accuracy) used in Table 2 and 4, we first find out the agent that corresponds to the column with the largest weight for each row ("column agent") of the adjacency matrices. Then, we construct an edge from the agent that corresponds to the row to the "column agent" then compare this with the ground truth graph. This comparison is fair to the baselines because we are constructing the same amount of edges for all models (i.e. one outbound edge for each agent). Note that ground truth graphs exist only for the Social Navigation Environment and PHASE. To avoid complications in calculating the mutual information betweeen two continuous vectors for the ablation study, we adopt the following procedure to calculate a "discrete" NMI score (used in Table 3): (1) label each element in the graph by their ordering within the row (e.g. [0.1, 0.9, 0.4, 0.5, 0.3] is transformed into [5, 1, 3, 2, 4]) and (2) calculate the averaged normalized mutual information between rows in graph one and rows in graph two. Intuitively, this metric measures "how much information does knowing about the first graph tell me about the second graph?"

## D.2   Relative Zero-shot Generalization Performance

Table 5: Zero-shot generalization relative to performance on original Social Navigation Environment. The top-most two metrics demonstrate that the change in performance of IMMA on generalization scenarios relative to the original environment is similar, and in many cases better, than the baselines; lower numbers are better, as they mean that the performance dropped less. The bottom-most four metrics demonstrate that IMMA (MG+PLT) outperforms the baselines by a similar, and in many cases better, percentage across generalization scenarios; higher numbers indicate better IMMA generalization performance, as they mean that IMMA (MG+PLT) more outperforms that baseline on the generalization scenario, relative to on the original environment.

| Metrics | Baseline Methods | | | | | | | IMMA (Ours) | | |
| | MLP | GAT_LSTM [41] | NRI [22] | EvolveGraph [27] | RFM (skip 1) | RFM [39] | fNRI [43] | SG | MG | MG + PLT |
|---|---|---|---|---|---|---|---|---|---|---|
| Mean drop from original performance (across generalization scenarios) | | | | | | | | | | |
| ADE | - | 0.41 | 0.12 | 0.08 | 0.11 | 0.07 | 0.07 | 0.03 | 0.03 | 0.04 |
| FDE | - | 0.71 | 0.27 | 0.25 | 0.18 | 0.18 | 0.15 | 0.06 | 0.07 | 0.08 |
| Graph Acc (%) | - | 0.56 | 8.23 | 6.92 | 7.57 | 7.34 | 4.92 | 6.62 | 5.69 | 6.56 |
| Mean % drop from original performance (across generalization scenarios) | | | | | | | | | | |
| ADE | - | 31 | 26 | 26 | 28 | 24 | 26 | 14 | 15 | 19 |
| FDE | - | 32 | 28 | 30 | 26 | 26 | 26 | 14 | 16 | 19 |
| Graph Acc (%) | - | 3 | 14 | 10 | 11 | 10 | 14 | 8 | 7 | 8 |
| % relative performance vs. MG+PLT (on original environment) / Mean % relative performance vs. MG+PLT (across generalization scenarios) | | | | | | | | | | |
| ADE | - | 1.22 | 1.11 | 1.12 | 1.15 | 1.09 | 1.11 | 0.94 | 0.96 | 1.00 |
| FDE | - | 1.23 | 1.15 | 1.18 | 1.11 | 1.11 | 1.11 | 0.93 | 0.96 | 1.00 |
| Graph Acc (%) | - | 0.94 | 1.09 | 1.02 | 1.04 | 1.03 | 1.09 | 1.00 | 0.99 | 1.00 |
| % relative performance vs. MG+PLT (original environment) / % relative performance vs. MG+PLT (2x speed) | | | | | | | | | | |
| ADE | - | 0.85 | 0.90 | 0.99 | 0.95 | 0.95 | 0.99 | 0.94 | 0.96 | 1.00 |
| FDE | - | 0.88 | 0.92 | 0.96 | 0.94 | 0.94 | 0.97 | 0.93 | 0.96 | 1.00 |
| Graph Acc (%) | - | 0.96 | 1.01 | 1.01 | 1.00 | 1.00 | 0.98 | 1.00 | 0.99 | 1.00 |
| % relative performance vs. MG+PLT (original environment) / % relative performance vs. MG+PLT (2x smaller) | | | | | | | | | | |
| ADE | - | 1.00 | 1.00 | 0.96 | 0.97 | 0.96 | 1.00 | 0.96 | 0.94 | 1.00 |
| FDE | - | 1.00 | 1.00 | 0.98 | 0.96 | 0.96 | 1.00 | 0.95 | 0.94 | 1.00 |
| Graph Acc (%) | - | 0.99 | 1.00 | 0.97 | 1.00 | 1.00 | 1.00 | 0.99 | 1.00 | 1.00 |
| % relative performance vs. MG+PLT (original environment) / % relative performance vs. MG+PLT (2x agents) | | | | | | | | | | |
| ADE | - | 3.46 | 1.73 | 1.58 | 1.96 | 1.52 | 1.46 | 0.92 | 0.99 | 1.00 |
| FDE | - | 3.31 | 1.95 | 2.12 | 1.67 | 1.65 | 1.47 | 0.91 | 0.99 | 1.00 |
| Graph Acc (%) | - | 0.87 | 1.31 | 1.10 | 1.12 | 1.10 | 1.36 | 1.02 | 0.99 | 1.00 |

## D.3 Additional Qualitative Analysis

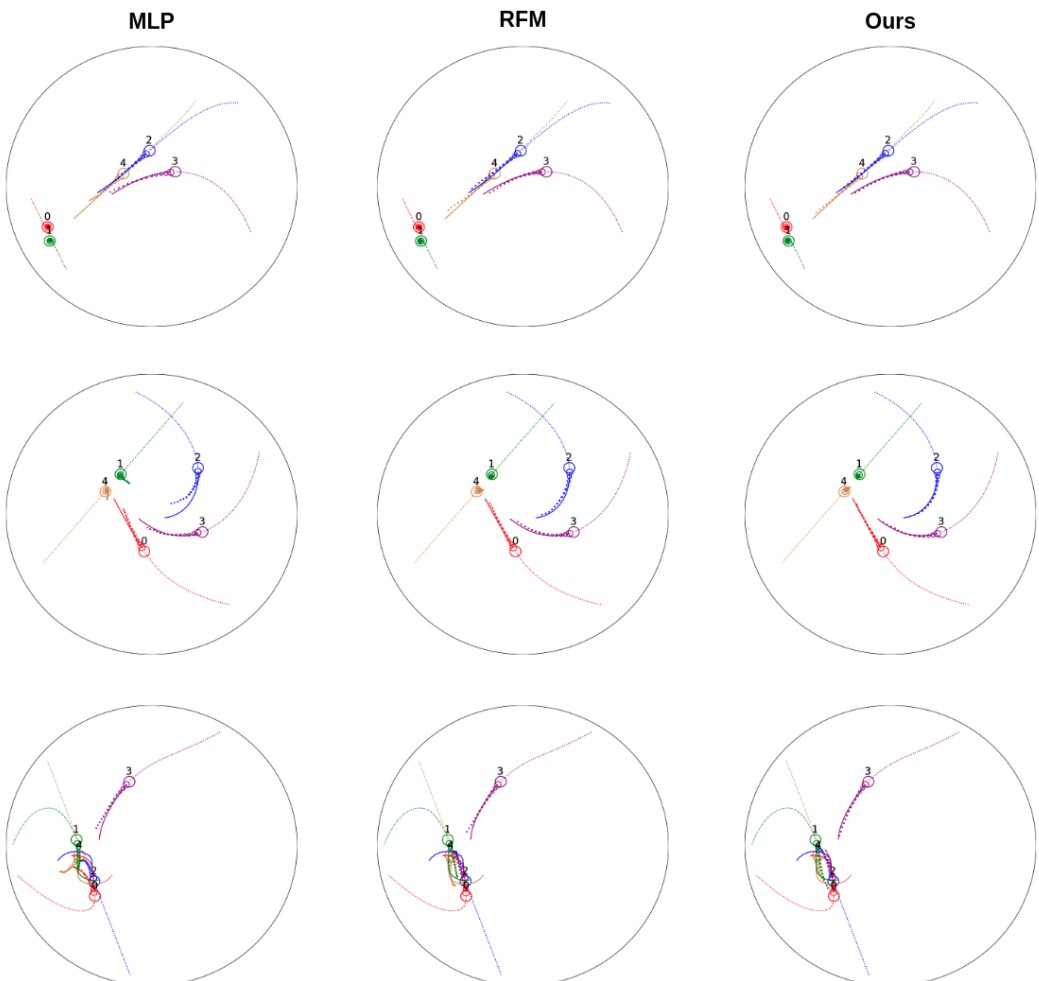

Figure 7: Prediction results of our model visualized on the Social Navigation Environment. Dotted lines: past trajectories. Solid lines: ground truth future trajectories. Circles: predicted future trajectories. Columns from left to right: prediction results of the MLP model, prediction results of RFM, prediction results of our model.

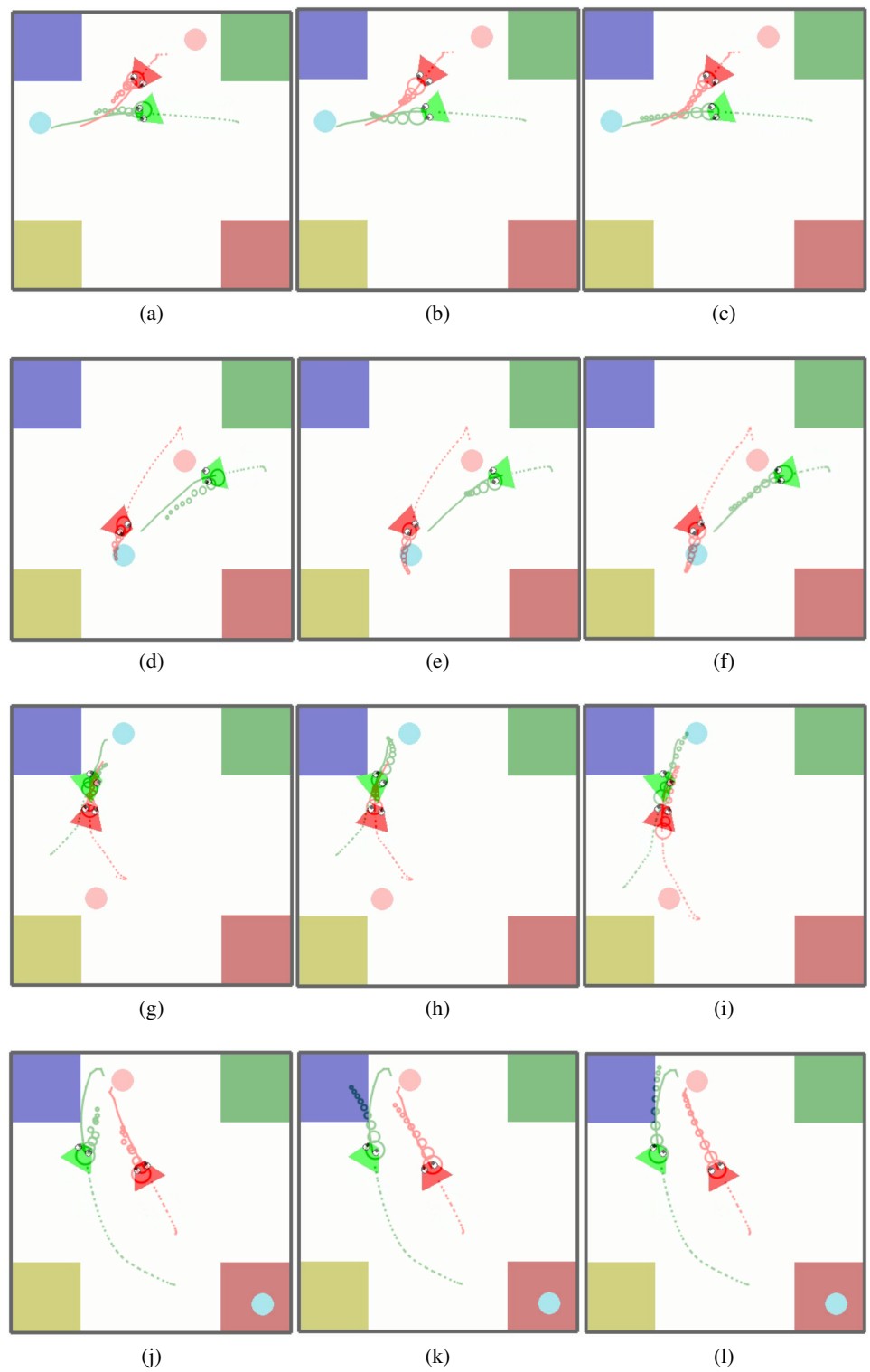

Figure 8: Prediction results visualized for PHASE collaboration task. Dotted lines: past trajectories. Solid lines: ground truth future trajectories. Circles: predicted future trajectories. Columns from left to right: prediction results of the MLP model, prediction results of RFM, prediction results of our model. Our model achieves the best performance in predicting future trajectories, even though it is far from perfect given the size of dataset (752 training samples before data augmentation).

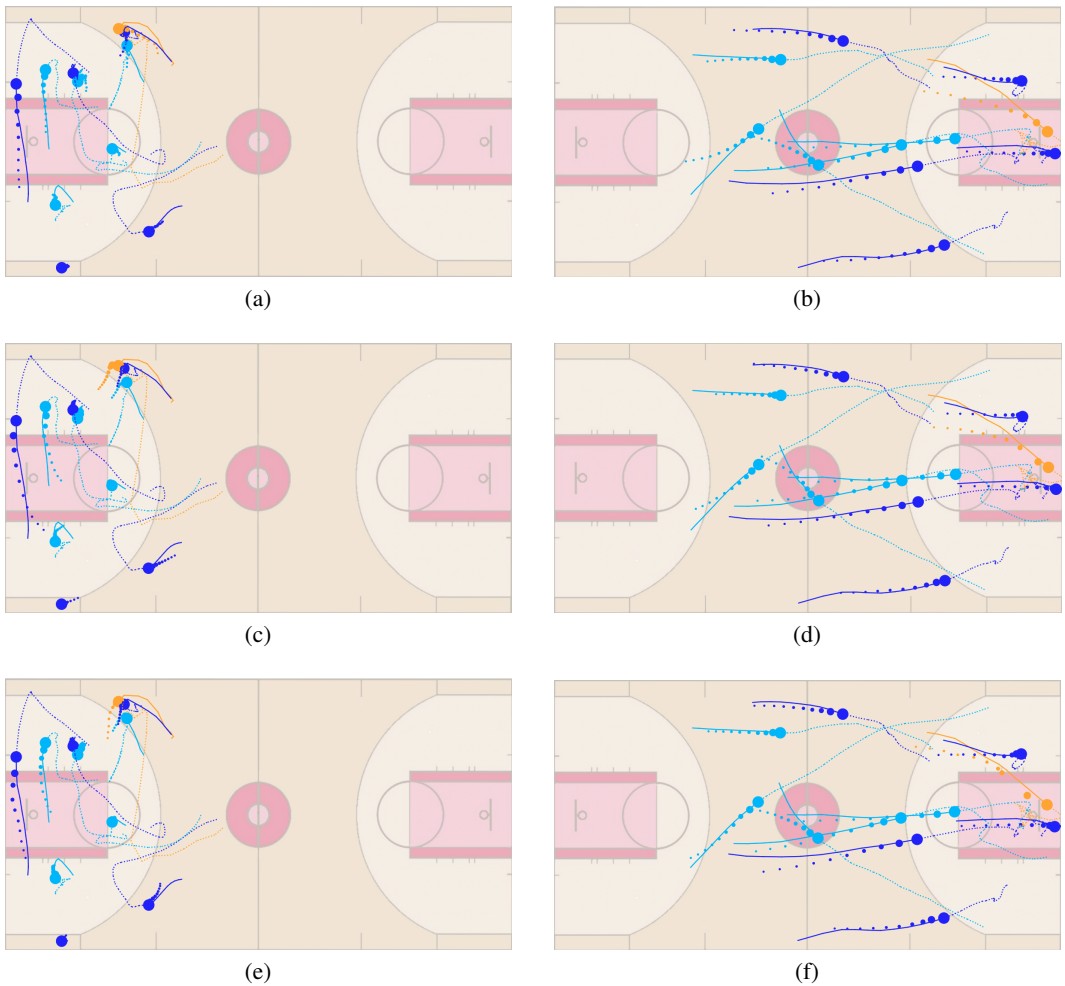

(a)                 (b)

(c)                 (d)

(e)                 (f)

Figure 9: Prediction results of the MLP model on the NBA dataset visualized. Dotted lines: past trajectories. Solid lines: ground truth future trajectories. Circles: predicted future trajectories. Rows from up to down: prediction results of the MLP model, prediction results of RFM, prediction results of our model.

**Before**       **After**

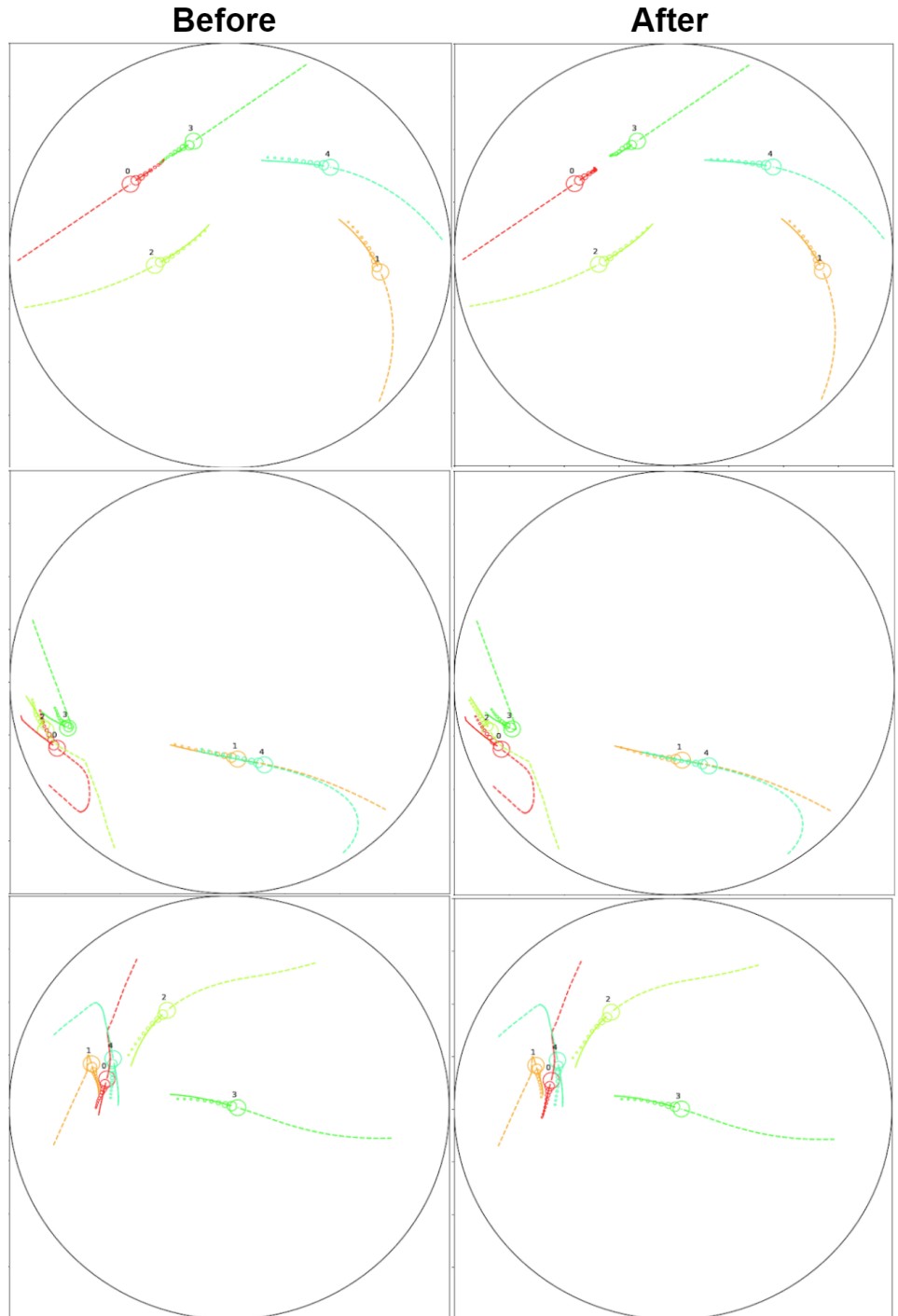

Figure 10: Prediction results of our model visualized on the Social Navigation Environment before and after the second latent graph layer is learned. Dotted lines: past trajectories. Solid lines: ground truth future trajectories. Circles: predicted future trajectories. We can see that after the second latent graph layer, the model learns to predict collision avoidance much better. For the first row, observe the predicted trajectories of the green and red agents. For the second row, observe the predicted trajectories of the red agent. For the third row, observe the predicted trajectories of the orange and red agents.

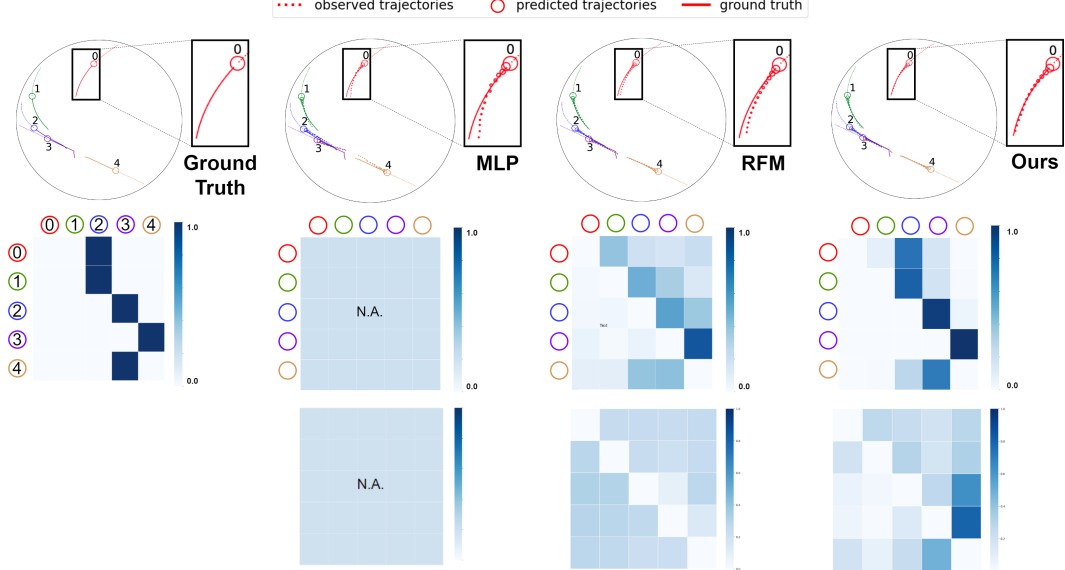

Figure 11: Visualization of the latent graph and agent trajectories of the Social Navigation Environment. (top) Predicted trajectories. The smaller the circle, the further it is into the future. The leftmost column shows ground truth trajectories and the ground truth graph used to simulate those trajectories. (bottom) Inferred latent graphs. The second row shows the first latent graphs and the last row shows the second latent graphs. Edge strength between agent $i$ and $j$ is represented by darkness of the cell at row $i$ and column $j$. The red agent's relational prediction is inaccurate with RFM—in the second row of the primary matrix, the green agent is incorrectly given higher weight than the blue agent—and thus the predicted trajectories deviate from the ground truth, especially on long-horizon predictions. This is essentially Figure 3 but with the second latent layer visualized, demonstrating that the second layer captures different interactions than the first layer.