# OpenReview forum: "Interaction Modeling with Multiplex Attention"
_NeurIPS.cc/2022/Conference — NeurIPS 2022 Accept_

### Official Review · Reviewer_FJKV · 2022-07-04

**Rating:** 7
**Confidence:** 3
**Soundness:** 3 good
**Presentation:** 3 good
**Contribution:** 3 good

**Summary:**

This paper presents a new method for modeling and predicting multi-agent interaction. Observing that such interactions could be affected by multiple factors such as target destinations and collision avoidance during crowd navigation tasks, the key idea was to model them using multiplex graphs. Specifically, rather than classifying types of edges of graphs as done in existing work, the proposed approach instead perform multi-class predictions for multiple edge types and construct multiplex attention matrices. To make latent graph layers diverse, the proposed work introduces a progressive learning scheme (progressive layer training) that gradually increases the number of graph layers to learn. Experimental results on multiple datasets show the effectiveness of the proposed approach over existing methods.


**Questions:**

Some suggestions:

- Figure 1 could be improved to precisely describe what the proposed method actually learns. Does it really learn to model collision avoidance, or does it just learn who follows whom?
- Honestly, I can see few differences between RFM and Ours in Figure 5 or cannot tell which of results from RFM and those from Ours are more natural.
- What are the limitations and failure cases for the proposed method?

**Limitations:**

As also described above, the paper currently has few discussions on the limitations and typical failure cases for the proposed approach.

**Strengths And Weaknesses:**

## Strengths
- **Originality**: modeling multi-agent interaction using multiplex graphs seems novel, at least for trajectory forecasting tasks. It is well motivated by the observations that human interactions can be affected by multiple factors.
- **Quality**: I agree that the proposed approach is well designed and technically sound. The questions that the paper addresses are clearly presented (Q1-Q4 in Section 4) and answered in the experimental results. Experiment details as well as potential societal impacts are clearly described in the appendix.
- **Clarity**: Overall the paper is clearly written and easy to follow.
- **Significance**: I believe that the proposed work is relatively significant mainly because it outperformed existing methods on multiple datasets. The zero-shot generalization experiment was interesting and showing the usefulness of the proposed approach.

## Weaknesses
- The motivating example for multiple factors affecting interactions, i.e., target friend and collision avoidance in Figure 1, may be a bit confusing. If I understand it correctly, multiplex graphs are built rather in a bottom-up fashion in the proposed method, where we cannot ensure that each graph layer has such semantic meaning. In fact, I'm still a bit confused what the multiplex graph actually learned. Is this just a leading agent as shown in Figure 5 or something beyond?
- While it was great that the proposed method performs very well, I would also like to see typical failure cases. Are there some cases where interaction modeling or prediction are likely to fail?

---

> ### Author Response · Authors · 2022-08-02
> **Initial response to reviewer FJKV (2/2)**
>
> - **Limitations and failure cases**
>
> We expect that our model would not be suitable for modeling dynamic systems where agents' relations are rapidly evolving. In our model, we use the same (multiplex) graph for all decoding steps, but that could be suboptimal when the underlying relations between agents change within the timespan of our forecasting horizon. We leave the incorporation of techniques available for dealing with this (e.g. [1,2]) to future work.
>
> Additionally, our model scales as N^2 in the number of agents, and thus may not be well-suited for scaling to large numbers of agents. Future work may be able to address this problem through incorporation of heuristics or priors about which interactions can be ignored from the beginning.
>
> [1] Graber, Colin, and Alexander Schwing. "Dynamic neural relational inference for forecasting trajectories." Proceedings of the IEEE/CVF Conference on Computer Vision and Pattern Recognition Workshops. 2020.
>
> [2] Li, Jiachen, et al. "Evolvegraph: Multi-agent trajectory prediction with dynamic relational reasoning." Advances in neural information processing systems 33 (2020): 19783-19794.

---

> > ### Comment · Reviewer_FJKV · 2022-08-07
> > **Thank you**
> >
> > Thank you very much for the detailed response. I have checked the comment as well as the other reviews and confirmed my rating (7: accept).
> > I think the discussion of limitations posted above is very important and should be clearly stated in the paper with a sub-section. It will make it easier for the community to further extend this work.

---

> > > ### Author Response · Authors · 2022-08-08
> > > **Thank you!**
> > >
> > > Thank you for taking the time and care to consider both our original manuscript and our responses. We will include a discussion of the limitations of our method in the revised version.

---

> ### Author Response · Authors · 2022-08-02
> **Initial response to  reviewer FJKV (1/2)**
>
> We thank the reviewer for the time and effort to review our manuscript and provide helpful feedback. In the following, we address the raised concerns point by point:
>
> - **Figure 1 as description of what the model learns**
>
> Figure 1 does describe our understanding of what the proposed method learns, in the context in which there is a dominant interaction (e.g. each follower wants to move towards a leader) and a local interaction (e.g. collision avoidance): who follows whom, as well as additional interactions like collision avoidance. We have added a figure (Appendix Figure 11) that visualizes multiple layers of the latent graph (as opposed to just the primary layer).
>
> - **What the multiplex graph has actually learned**
>
> In our method, latent codes are multiplex graphs, represented as adjacency matrices that describe interactions between agents. The primary graph is the first graph learned in Progressive Layered Training and should encode the primary interactive relationship that could reduce prediction loss the most. In interpreting, leaders are identified by choosing the largest entry for each column agent of the primary graph.
>
> IMMA learns these graphs purely based on the objective of minimizing prediction loss, and thus we do not know for certain the semantic meaning of each layer. However, we mainly evaluate the primary graph, through qualitative visualization (Figure 3 and Figure 5) and quantitive assessment (Table 2), because we have strong intuition as to what the first graph should encode, given the nature of the environment. The primary graph is the first graph learned in the process of progressive layered training and should encode the primary interactive relationship that could reduce prediction loss the most. For instance, in the Social Navigation Environment and the PHASE dataset, the primary graph should learn to encode the “leader-follower” relations since such knowledge would reduce prediction loss the most. As shown in figure 3 and figure 5, the primary graph learns the leader-follower relationship – which aligns with our understanding of the environment. In the main text, we only visualized the primary graph (i.e., the first graph learned during progressive layered training) because we have strong intuitions about what the first layer should encode.
>
> To further support our claim that different graph layers can indeed learn to encode different types of interactions, we updated the supplementary materials (Appendix Figures 10 and 11) to include two additional qualitative studies: (a) visualization of the second latent graph in the supplementary material, and (b) comparison of predictions before and after learning the second graph. The results showed that the second graph layer seems to learn collision-relevant interactions, and importantly, that incorporation of the second latent graph did help with modeling collision avoidance behaviors.
> - **Differences between RFM and Ours with conditional generation (Fig 5.)**
>
>
> Regarding analysis of the conditional generation experiments, achieved through latent graph manipulation, we want to preface our answer by saying that we do not have the ground truth for this analysis. This is a qualitative analysis, and we evaluate the predictions based on our understanding of the environment. We claim that the generated trajectories of the baseline are less optimal because (a) it consists of unnatural turns (i.e., when we change the leader of agent 0 to 1), and (b) other agents’ leaders remain the same, and thus their trajectories should not be affected when we only change the leader of agent 0. For example, agents 3 and 4 follow each other, and their behavior should not change throughout the process, yet the baseline method predicts that they would make a turn.
>
> To quantify this evaluation, we ran a two-alternative forced (2AFC) choice human study with 20 subjects, using the trajectories shown in Fig. 5, in which subjects were asked to choose which model’s prediction is more reasonable (RFM vs. ours). In each of the three trials, the order of the predictions is randomized and the subjects do not have access to which prediction is generated by which model. The results show that our model’s prediction is judged as more reasonable than the RFM model’s in 76.67% of the trials.

---

### Official Review · Reviewer_PvS1 · 2022-07-09

**Rating:** 6
**Confidence:** 3
**Soundness:** 3 good
**Presentation:** 3 good
**Contribution:** 2 fair

**Summary:**

The authors proposed a prediction model that uses a multiplex latent graph to represent multiple types of interactions and attention to account for different relations called Interaction Modeling with Multiplex Attention (IMMA). They also introduce a training strategy called Progressive Layer Training (PLT). The experimental results showed that their approach outperformed baseline models in trajectory forecasting and relation inference, spanning three multi-agent scenarios: social navigation, cooperative task achievement, and team sports. They also demonstrated better zero-shot generalization results in the social navigation environment.



**Questions:**

1. I don’t know all of the trajectory prediction studies, but multilayer graph neural networks in trajectory prediction may not be a new idea and the following studies were quickly found [1] [2].
2. The idea of PLT may seem to be a special case of the original paper (Li et al. 2020 ICLR, not arxiv). The authors can clarify the difference from the original idea.
3. The strength of the method may be the decomposition of multilayer relations like Fig.1. However, the results including Fig 3 may show only one layer. How did the authors merge it? And I want to know whether each layer can extract multiple relations like Fig. 1 or not.

[1] Inhwan Bae and Hae-Gon Jeon, Disentangled Multi-Relational Graph Convolutional Network for Pedestrian Trajectory Prediction, AAAI, 2021
[2] Rui Zhou et al. Grouptron: Dynamic Multi-Scale Graph Convolutional Networks for Group-Aware Dense Crowd Trajectory Forecasting, ICRA 2022


**Limitations:**

The authors addressed the limitations and potential negative societal impact of their work.

**Strengths And Weaknesses:**

Stremgth:
* They proposed a prediction model that combines multilayer graph neural networks and PLT for interpreting the multiplexed latent structure.
* The experiments showed that their approach outperformed baseline models in trajectory forecasting and relation inference, spanning three multi-agent scenarios. They also demonstrated superior sample efficiency and better zero-shot generalization results.

Weakness:
* The novelty of this method seems to be combining multilayer graph neural networks and PLT, but the authors mentioned that each component has novelty, which may be exaggerated. I asked this question in the following “Questions”.
* The result was clear but there was no code available (I saw “Checklist”) and there were some questions in the experiment description.

---

> ### Author Response · Authors · 2022-08-02
> **Initial response to reviewer PvS1**
>
> We thank the reviewer for the time and effort to review our manuscript and provide insightful feedback. In the following, we address the raised concerns point by point:
>
> - **Difference from other multi-layer GNNs for trajectory prediction**
>
> Although there are similarities to related methods, our work is novel both in terms of the individual components of the method as well as the way in which we combine them.
>
> Compared to existing works that use multi-layer GNNs [3,4] our work is novel for the following reasons:
>
> (1)  Existing works use different layers of graphs to explicitly model interactions of different scales [3,4], whereas we train our multiplex graphs with Progressive Layered Training to let the model automatically learn what to encode for each layer.
>
> (2) Existing works treat the graphs as a transient part of the computation and did not conduct quantitative analysis on the relations constructed. On the other hand, the graph is treated as an integral part of the computation in IMMA, and we go further by evaluating the learned relations quantitatively and qualitatively. This conceptual difference is also discussed in section 3 of the NRI paper [5].
>
> (3) Existing works simply use weighted graphs. We propose using attentional graphs as the latent structure because we believe it is essential to model relative strengths in social dynamical environments. Our experiments showed that using attentional graphs boosts trajectory prediction performance and interpretability.
>
> - **Difference from Progressive Learning (Li et al. 2020 ICLR)**
>
> The proposed Progressive Layered Training (PLT), utilizing progressive training on a layered latent space as a curriculum learning technique, is novel to the best of our knowledge (progressive training [1] was first proposed to solve the problem of catastrophic forgetting). We conducted ablation studies (table 3) to provide insights as to why this training technique is crucial – finding that it helps with the disentanglement between different interaction types (i.e., graph layers).
>
> PLT is related to [2], but fundamentally different. They use progressive learning to learn hierarchical representations (i.e., lower-level latent codes are results of a function that takes in higher-level representations as input) to improve disentanglement. On the other hand, we can describe progressive layered training as modeling residuals by different neural networks, and each successive layer simply adds accuracy to an already-functional but less accurate predictor. Additionally, our main intention is to improve trajectory prediction performance, and improved disentanglement is a byproduct that appears to explain the effectiveness of PLT.
>
> - **Visualizing layers of relations**
>
> For visualization, we did not merge the layers of relations. In the main text, we only visualized the primary graph (i.e., the first graph learned during progressive layered training) because we have strong intuitions about what the first layer should encode.
> To further support our claim that different graph layers can indeed learn to encode different types of interactions (as schematized in Figure 1), we updated the supplementary materials to include two additional qualitative studies: (a) visualization of the second latent graph in the supplementary material, and (b) the predictions before and after learning the second graph. The results showed that incorporation of the second latent graph did help with modeling collision avoidance behaviors.
>
> - **Code availability**
>
> The code of our model is available here: https://drive.google.com/file/d/1h7o6xJp6odsJ6RvZxzT30w0n_V-lDkGh/view?usp=sharing. We will release the entire repository upon publication.
>
> [1] Rusu, Andrei A., et al. "Progressive neural networks." arXiv preprint arXiv:1606.04671 (2016).
>
> [2] Li, Zhiyuan, et al. "Progressive learning and disentanglement of hierarchical representations." arXiv preprint arXiv:2002.10549 (2020).
> ICLR version: https://openreview.net/forum?id=SJxpsxrYPS
>
> [3] Inhwan Bae and Hae-Gon Jeon, Disentangled Multi-Relational Graph Convolutional Network for Pedestrian Trajectory Prediction, AAAI, 2021
>
> [4] Rui Zhou et al. Grouptron: Dynamic Multi-Scale Graph Convolutional Networks for Group-Aware Dense Crowd Trajectory Forecasting, ICRA 2022
>
> [5] Kipf, Thomas, et al. "Neural relational inference for interacting systems." International Conference on Machine Learning. PMLR, 2018.

---

> > ### Comment · Reviewer_PvS1 · 2022-08-09
> > **Thanks for the rebuttal**
> >
> > Thank you for the rebuttal.
> > I understood your responses other than the third one.
> > For visualizing layers of relations, I saw Figures 10 and 11. I understood Figure 10, but I cannot interpret the second layer in Figure 11 (right bottom).
> > Totally, my unclear points are clarified, but probably due to the combination of the methodological minor changes, I did not change the score.

---

### Official Review · Reviewer_diNz · 2022-07-10

**Rating:** 6
**Confidence:** 4
**Soundness:** 3 good
**Presentation:** 3 good
**Contribution:** 3 good

**Summary:**

The paper proposes a multiplex attention method for multi-agent interaction modeling. The main motivation is that there can be more than one interaction type between agents. To address this, the paper proposes to use multiple latent graphs to encode the different interactions between agents and employs the VAE framework to infer the latent codes. To ease learning, the paper also proposes progressive layer training, which is a curriculum learning method that gradually adds latent graphs to the model. Evaluations are performed on multi-agent trajectory forecasting using one in-house simulated dataset and two public datasets, PHASE and NBA. The method outperforms prior methods in forecasting accuracy, relational inference accuracy, and sample efficiency.

**Questions:**

- In Fig. 3, how are the latent codes identified for changing the leader? If I understand correctly, there should be multiple latent codes. A more compelling experiment for interpretability is probably to show different latent codes can lead to different types of interactions (besides leader-follower). This also validates the key motivation of this paper better.
- Are latent edges directional? i.e., is $z_{ij}$ different than $z_{ji}$? Eq (2) only makes sense if edges are directional, otherwise $i$ and $j$ are not symmetric when computing $z_{ij}$.
- Zero-shot generalization experiments are not convincing in my opinion. The performance of all methods has dropped but in some cases the performance gap between the current method and the baselines becomes smaller. Normally, the method should outperform the baselines by a similar percentage in this new setting. Doesn’t this indicate the method actually generalizes worse than some of the baselines?
- How are column agents selected for the relational inference in Table 2? Since the method has multiple latent graphs, do you choose the largest latent code as the latent code for each column agent?



**Limitations:**

- The paper didn’t really talk about its limitation despite indicating in the form it has done so.
- One clear limitation is that it cannot handle changing interaction types. As the multi-agent interaction evolves over time, their interaction could also evolve. For example, a follower passes a leader to become the new leader. Recent work on dynamic NRI [5, 6] has attempted to address this limitation.

[5] Graber, Colin, and Alexander Schwing. "Dynamic neural relational inference for forecasting trajectories." Proceedings of the IEEE/CVF Conference on Computer Vision and Pattern Recognition Workshops. 2020.

[6] Xiao, Ruichao, Manish Kumar Singh, and Rose Yu. "Dynamic Relational Inference in Multi-Agent Trajectories." ICLR 2021.

**Strengths And Weaknesses:**

**Strength:**

- The paper is generally well-written and ways to understand. The motivation is clear for encoding multiple types of interaction between agents and their relative strength.
- The paper performed extensive experiments and analyses on three datasets. Both trajectory forecasting and relation inference are evaluated. The latent graph manipulation experiments are interesting which show the latent codes encode leader-follower behaviors.
- The paper outperforms the baselines significantly.

**Weakness:**

- The main idea, multiplex attention, of this paper is quite **similar to factorized NRI [1]**, where multiple latent graphs are used in a variational inference framework. The technical novelty of this paper is limited. In some sense, the multiplex attention is not a big change from NRI since the latent code itself is multi-dimensional (therefore each dimension can be treated as a separate latent code) and the main difference of the proposed approach is to not share encoder weights.
- The paper didn’t compare with strong multi-agent trajectory forecasting baselines on classic trajectory forecasting benchmarks such as ETH/UCY where a large number of agents are interacting with each other. The paper could compare with SOTA methods such as [2, 3, 4].
- Some design choices are not well motivated. What is the motivation for soft-max across agents in Eq (2)? This constrains the interaction strength from $i$ to other agents to sum to 1 for interaction $k$. However, agent $i$ can have strong interaction strength with multiple different agents for interaction $k$. Therefore, I wonder why the method doesn’t just formulate inference of $z_{ij}^k$ as a binary classification (this interaction exists or not) instead of multi-class classification across other agents.

[1] Webb, Ezra, et al. "Factorised neural relational inference for multi-interaction systems." ICML 2019 Workshops.

[2] Mangalam, Karttikeya, et al. "From goals, waypoints & paths to long term human trajectory forecasting." Proceedings of the IEEE/CVF International Conference on Computer Vision. 2021.

[3] Yuan, Ye, et al. "Agentformer: Agent-aware transformers for socio-temporal multi-agent forecasting." Proceedings of the IEEE/CVF International Conference on Computer Vision. 2021.

[4] Wang, Chuhua, et al. "Stepwise goal-driven networks for trajectory prediction." IEEE Robotics and Automation Letters 7.2 (2022): 2716-2723.

---

> ### Author Response · Authors · 2022-08-02
> **Initial response to reviewer diNz (3/3)**
>
> - **Relative zero-shot generalization performance**
>
> Further analysis of our generalization results shows that when considering the variety of metrics (ADE/FDE/graph accuracy) and experiments (2x speed, 2x size, 2x agents), IMMA shows a better change in performance when averaged across the three generalization scenarios (Appendix Table 5). In particular, IMMA has a drop in ADE of 0.03 to 0.04, versus a baseline drop of 0.07 to 0.41. Similarly, IMMA has a drop in FDE of 0.06 to 0.08 versus a baseline drop of 0.15 to 0.71.  With Graph Accuracy, where there was a large spread in the original (non-generalizing) performance of methods, if we disregard the two methods that performed very poorly on this metric in the original environment (fNRI and GAT_LSTM), we see that IMMA dropped less than the other high-performing baseline methods (NRI, EvolveGraph, RFM and RFM_skip1), with a drop of ~6 versus ~7 for the baseline methods. Similar results hold when using percent change in performance. Additionally, when averaged across generalization scenarios, IMMA also outperforms the other methods by a greater percentage, and a similar or greater percentage on each generalization scenario.We are grateful for the suggestion to represent the generalization experiment metrics in this way and have included a supplemental table that reports this directly.
>
> Additionally, the fact that our model outperforms other models in absolute terms is meaningful: it implies that our model does not just find a way to overfit the current environment and that it learns to infer more accurate relations in a way that is translatable to moderately out-of-distribution scenarios.
>
> - **Relational inference from latent code**
>
> We chose the largest entry for each column agent of the primary graph. The primary graph is the first graph trained according to our progressive training process.
>
> - **Discussion of limitations**
>
> We believe our model would not be suitable for modeling dynamic systems where agents' relations are rapidly evolving. In our model, we use the same (multiplex) graph for all decoding steps, but that could be suboptimal when the underlying relations between agents change within the timespan of our forecasting horizon. We leave the incorporation of techniques available for dealing with this (e.g. [6, 7]) to future work.
>
> Additionally, our model scales as N^2 in the number of agents, and thus may not be well-suited for scaling to large numbers of agents. Future work may be able to address this problem through incorporation of heuristics or priors about which interactions can be ignored from the beginning.
>
> [6] Graber, Colin, and Alexander Schwing. "Dynamic neural relational inference for forecasting trajectories." Proceedings of the IEEE/CVF Conference on Computer Vision and Pattern Recognition Workshops. 2020.
>
> [7] Li, Jiachen, et al. "Evolvegraph: Multi-agent trajectory prediction with dynamic relational reasoning." Advances in neural information processing systems 33 (2020): 19783-19794.

---

> > ### Comment · Reviewer_diNz · 2022-08-07
> > **Thanks for the rebuttal**
> >
> > The rebuttal has addressed my main concerns and I'm happy to increase my rating to weak accept. I still encourage the authors to compare again traditional trajectory forecasting methods since the two fields, interaction modeling/relational inference & trajectory forecasting, are working on similar problems but often do not compare with each other. It would be helpful to integrate the efforts and measure the progress together.

---

> > > ### Author Response · Authors · 2022-08-08
> > > **Thank you!**
> > >
> > > Thank you for taking the time and care to consider both our original manuscript and our responses. This has certainly made the work stronger and clearer.

---

> ### Author Response · Authors · 2022-08-02
> **Initial response to reviewer diNz (2/3)**
>
> - **Selection of baselines for comparison**
>
> We chose to compare with NRI, RFM, and EvolveGraph because they are considered state-of-the-art in the line of works that explicitly learn an interaction graph as part of the trajectory prediction task. That is, our chosen baselines not only learn an interaction graph but also explicitly evaluate the interpretability of this graph, whereas many other trajectory prediction works  [2,3,4] treat latent codes as a transient part of the computation. This difference is also discussed in the NRI paper [5]. Our paper falls into this line of research.
> We prioritize comparing our model with EvolveGraph among the “SOTA models’’ because it is one of the few that has shown success in highly interactive environments such as the NBA dataset, that has been evaluated on relational inference, and that was also compared with trajectory prediction-focused models such as Trajectron++. Most other multi-agent trajectory forecasting works [2,3,4] mainly experiment with pedestrian prediction datasets (e.g., ETH/UCY) or autonomous vehicle trajectory datasets (e.g., nuScenes, drone datasets), for which modeling interactions is less important [1]. Thus, among the “SOTA methods” in the “traditional” multi-agent trajectory prediction works, we deem EvolveGraph the best method for us to compare with.
>
> - **Motivation for soft-max across agents**
>
> We propose to use attention (i.e., softmax across agents) for each layered graph in the multiplex graph so that the relative strengths of interactions can be modeled more naturally. Note that we do not argue multiplex attention is the better formulation for all interactive systems, but our insight is that it is important for modeling relative strengths in social dynamical environments. Aside from yielding great performance, using attention provides the two following advantages: (1) increased interpretability (i.e., the ability to answer “what causes an agent to behave this way?”), as seen in table 2, and (2) the ability to be directly applicable to inductive learning problems as the softmax in the attention mechanism naturally normalizes the weights across all agents. This makes it more amenable to arbitrary agent counts, which is supported by our experimental results in table 4. Observe that our model with attention significantly outperforms other models in the zero-shot generalization experiment, where we incorporate twice the number of agents into the environment.
>
> - **Interpretation of latent codes, and Identification of codes for changing the leader**
>
> In our method, latent codes are multiplex graphs, represented as adjacency matrices that describe interactions between agents. The primary graph is the first graph learned in Progressive Layered Training and should encode the primary interactive relationship that could reduce prediction loss the most.  Leaders are identified by choosing the largest entry for each column agent of the primary graph. To change the leader of a specific agent, we go to the row index of the agent and simply assign 1 for the leader column and 0 for all other agents in the primary graph.
>
> IMMA learns these graphs purely based on the objective of minimizing prediction loss, and thus we do not know for certain the semantic meaning of each layer. However, we evaluate mainly the primary graph, through qualitative visualization (Figure 3 and Figure 5) and quantitative assessment (Table 2), because we have strong intuition as to what the first graph should encode, given the nature of the environment. For instance, in the Social Navigation Environment the primary graph should learn to encode the “leader-follower” relations since such knowledge would reduce prediction loss the most. As shown in Figure 3 and Figure 5, the primary graph learns the leader-follower relationship – which aligns with our understanding of the environment.
>
> - **Latent edges are directional**
>
> The latent edges are directional, as $z_{ij}$ is different than $z_{ji}$. We have updated the manuscript to make this more clear.
>
> [1] Makansi, Osama, et al. "You mostly walk alone: Analyzing feature attribution in trajectory prediction." arXiv preprint arXiv:2110.05304 (2021).
>
> [2] Mangalam, Karttikeya, et al. "From goals, waypoints & paths to long term human trajectory forecasting." Proceedings of the IEEE/CVF International Conference on Computer Vision. 2021.
>
> [3] Yuan, Ye, et al. "Agentformer: Agent-aware transformers for socio-temporal multi-agent forecasting." Proceedings of the IEEE/CVF International Conference on Computer Vision. 2021.
>
> [4] Wang, Chuhua, et al. "Stepwise goal-driven networks for trajectory prediction." IEEE Robotics and Automation Letters 7.2 (2022): 2716-2
>
> [5] Kipf, Thomas, et al. "Neural relational inference for interacting systems." International Conference on Machine Learning. PMLR, 2018.

---

> ### Author Response · Authors · 2022-08-02
> **Initial response to reviewer diNz (1/3)**
>
> We thank the reviewer for their thorough reading of our work and insightful suggestions for improvement. We have updated the manuscript in reaction to these suggestions, which we hope considerably strengthens it. Here, we address the concerns point by point:
>
> - **Differences from factorized NRI**
>
> We have now included fNRI (i.e., each layer-graph effectively only contains a single edge-type and sigmoid activation is used) as a baseline and have updated the manuscript with the new results. While fNRI is a competitive baseline, our model still outperforms in both trajectory prediction and relational inference.
>
> With respect to factorized NRI (fNRI), our approach has a number of differences:
>
> (1) fNRI proposed use of a multiplex graph where each layer of the graph is a binary classification problem. They proposed to use sigmoid as an alternative to the formulation used in NRI. In our paper, we proposed to use attentional graphs (multi-class classification across agents) in the form of softmax, which we find to be important for modeling relative strengths in social dynamical environments. Please refer to the below question 'Motivation for soft-max across agents' for further explanation.
>
> (2) One of our innovations is to combine multiplex graph latent structure with progressive training. According to our ablation studies, this is a key component that makes our model excel (Table 3). We believe Progressive Layered Training is just as important of a contribution to our approach as the proposed model architecture.
>
> (3) fNRI only conducts experiments on spring datasets and has very limited results. We substantially expand the scope and application realm, and even if the fNRI work touches upon similar ideas, it should not preclude our contributions.
>
> - **Why did we not benchmark on ETH/UCY**
>
> Past literature [1] has pointed out that modeling interactions has an insignificant effect on trajectory prediction accuracy with ETH-UCY, in comparison to simply using the target agent's historical trajectories. More specifically, they evaluated ablated versions of multiple existing trajectory prediction models (i.e., Trajectron++, PECNet, Social-STGCNN) and found that incorporating the interaction modeling module does not meaningfully affect the trajectory prediction accuracy on pedestrian trajectory datasets such as ETH-UCY. However, substantial performance drops are observed on highly interactive datasets such as the NBA dataset when interactions are not modeled. Thus, in this paper, we focus on benchmarking our model on environments where interaction modeling is indispensable to making accurate future predictions. We compare IMMA with models that have demonstrated success in similar regimes.
>
> [1] Makansi, Osama, et al. "You mostly walk alone: Analyzing feature attribution in trajectory prediction." arXiv preprint arXiv:2110.05304 (2021).

---

### Official Review · Reviewer_Q2Zh · 2022-07-11

**Rating:** 6
**Confidence:** 3
**Soundness:** 3 good
**Presentation:** 3 good
**Contribution:** 3 good

**Summary:**

This work tackles the problem of modeling multi-agent systems that contain a high degree of interaction between agents. To model such systems, this work proposes Interaction Modeling with Multiplex Attention (IMMA), which uses a multiplex graph latent structure to model different types of interactions in different layers of the multiplex. In addition, attention graph layers are used to capture the strength of the relations. Furthermore, thanks to the structure of the multiplex latent layers, this work proposes progressive layer training (PLT) where one can train different interaction layers separately, and iteratively adding different latent layers. The proposed approach is evaluated on three different datasets showcasing strong results on all. Furthermore, ablation studies show how having multiple latent layers and/or having PLT help the model’s prediction performance. Furthermore, the Social Navigation Environment provides the ground-truth interaction structure of each scene. This work shows that the predicted latent structure predicts ground truth more accurately than prior approaches.

**Questions:**

- I’m wondering where the results of EvolveGraph and NRI are obtained. EvolveGraph reports minADE_20/minFDE_20, and use 5 historical timesteps (2.0s) to make the prediction over the next 4 seconds. Could the authors clarify if this was their own implementation?
- Furthermore, there are multiple other works that report results on the NBA dataset, following the same configuration mentioned in my previous questions. These include STAR[1], PECNet[2], NMMP[3], GroupNet[4]. How come comparisons were not performed against these ones?
- Related to my previous question, can IMMA produce multiple modes of predictions or is it always unimodal?
- I’m confused about the footnote (2) on page 3 which states that one can ignore the KL divergence term. I have looked through the provided reference and could not find why it is possible to ignore this term. Can the authors clarify this?
- Figure 5: it is not clear to me why the predictions produced by IMMA are better than the baseline when the latent graph is manipulated. - What makes the new generated turns unrealistic in the baseline when compared to IMMA’s predictions? I will say that it is an interesting feature that the predictions of the other agents do not change.

References:

[1] Yu, C., Ma, X., Ren, J., Zhao, H. and Yi, S., 2020, August. Spatio-temporal graph transformer networks for pedestrian trajectory prediction. In European Conference on Computer Vision (pp. 507-523). Springer, Cham.

[2] Mangalam, K., Girase, H., Agarwal, S., Lee, K.H., Adeli, E., Malik, J. and Gaidon, A., 2020, August. It is not the journey but the destination: Endpoint conditioned trajectory prediction. In European conference on computer vision (pp. 759-776). Springer, Cham.

[3] Hu, Y., Chen, S., Zhang, Y. and Gu, X., 2020. Collaborative motion prediction via neural motion message passing. In Proceedings of the IEEE/CVF conference on computer vision and pattern recognition (pp. 6319-6328).

[4] Xu, C., Li, M., Ni, Z., Zhang, Y. and Chen, S., 2022. GroupNet: Multiscale Hypergraph Neural Networks for Trajectory Prediction with Relational Reasoning. In Proceedings of the IEEE/CVF Conference on Computer Vision and Pattern Recognition (pp. 6498-6507).


**Limitations:**

These are adequately addressed in the conclusion and the appendix.

**Strengths And Weaknesses:**

Strengths:
- The paper is well organized and well-written.
- I believe the application of multiplex attention in the “latent space” is novel to the problem of multi-agent motion forecasting with interaction modeling.
- The results are strong and convincing on multiple datasets. The ablation studies provide further evidence of the utility of having multiple levels of attention.

Weaknesses:
- In the appendix, it is mentioned that training is performed with $q_\phi(z|X^{1:T_h})$. This, combined with the footnote on page 3 (on which I have a question below), it seems like the proposed approach should not be formulated/classified as a CVAE, and is more of a model which has a specific hidden structure on the bottleneck that is not regularized by any external forces (e.g., KL divergence).

---

> ### Author Response · Authors · 2022-08-02
> **Initial response to Reviewer Q2Zh (2/2)**
>
> - **Multiple modes of prediction by IMMA**
>
> There are two ways for IMMA to produce multiple modes of prediction. The first method is to treat every row in the latent graph as a discrete latent variable and sample different categorical latent codes to produce different modes of prediction (i.e., consider Gumbel-Softmax VAE [10]). Figure 5 shows predictions of our models with different latent codes, except that the latent codes are not sampled but handcrafted. The second way to produce multiple modes of predictions is to adapt our model to predict the means and variances of a Gaussian mixture distribution and sample from this multi-modal distribution during test time, similar to what EvolveGraph did.
>
> - **Improved predictions by IMMA with latent graph manipulation**
>
> We believe that the generated trajectories of the baseline are less realistic on two accounts: (a) they consist of unnatural turns (i.e., when we change the leader of agent 0 to 1), and (b) other agents’ leaders remain the same, and thus their trajectories should not be affected when we only change the leader of agent 0.  For example, agents 3 and 4 follow each other, and their behavior should not change throughout the process, yet the baseline method predicts that they would make a turn.
> Using the trajectories shown in Fig. 5, we ran a two-alternative forced (2AFC) choice human study with 20 subjects, in which subjects were asked to choose which model’s prediction is more reasonable (RFM vs. ours). In each of the three trials, the order of the predictions is randomized and the subjects do not have access to which prediction is generated by which model. The results show that our model’s prediction is judged as more reasonable than the RFM model’s in 76.67% of the trials.
>
> [10] Jang, Eric, Shixiang Gu, and Ben Poole. "Categorical reparameterization with gumbel-softmax." arXiv preprint arXiv:1611.01144 (2016).

---

> ### Author Response · Authors · 2022-08-02
> **Initial response to Reviewer Q2Zh (1/2)**
>
> We thank the reviewer for their time and effort reading our work and providing detailed and insightful feedback. In the following, we address the concerns point by point:
>
> - **Formulation of IMMA as a CVAE**
>
> We decided to frame our model as a CVAE because it allowed us to include a KL divergence term during training and to facilitate an understanding of how the proposed Multiplex Attention Graph might be thought of as an effective prior on the latent space.
> The KL divergence term acts as a prior (i.e., regularization) on the latent variables. If we drop the KL divergence term in the standard (conditional) VAE formulation (Equation 1), it reduces to an autoencoder. In our experiments, we found that incorporating the KL divergence term does not improve the performance of our model. We believe the explanation is that using Multiplex Attention Graph for the latent structure imposes a prior on the latent space. We are happy to adjust our writing if this would be better grasped by another description.
>
> - **Implementation of EvolveGraph and NRI baselines**
>
> For NRI, we used the official repository provided here (https://github.com/ethanfetaya/NRI). For EvolveGraph, we reached out to the authors and were able to use the official code for all our experiments.
>
> - **Selection of baselines for NBA dataset**
>
> We chose to compare with NRI, RFM, and EvolveGraph because they are considered state-of-the-art in the line of works that explicitly learn an interaction graph as part of the trajectory prediction task. Moreover, our chosen baselines not only learn an interaction graph but also explicitly evaluate the interpretability of this graph, whereas many other trajectory prediction works treat latent codes as a transient part of the computation [1,2,3,4,5]. This difference is also discussed in the NRI paper [6].
> Furthermore, many multi-agent trajectory forecasting works conduct experiments on pedestrian prediction datasets (e.g., ETH/UCY) or autonomous vehicle trajectory datasets (e.g., nuScenes, drone datasets), for which modeling interactions has been shown to be less important [7]. We prioritize comparing our model with EvolveGraph among the “SOTA models’’ because it has shown success in highly interactive environments such as the NBA dataset, has evaluated its model on relational inference, and has also been compared with many trajectory prediction-focused models such as Trajectron++ [8].
> Additionally, we included a new baseline fNRI [9], a model that also uses a multiplex latent graph structure (i.e., we consider the variant in which sigmoid activation is used) but does not use attention or progressive training.
>
> [1] Yuan, Ye, et al. "Agentformer: Agent-aware transformers for socio-temporal multi-agent forecasting." Proceedings of the IEEE/CVF International Conference on Computer Vision. 2021.
>
> [2] Yu, Cunjun, et al. "Spatio-temporal graph transformer networks for pedestrian trajectory prediction." European Conference on Computer Vision. Springer, Cham, 2020.
>
> [3] Mangalam, K., Girase, H., Agarwal, S., Lee, K.H., Adeli, E., Malik, J. and Gaidon, A., 2020, August. It is not the journey but the destination: Endpoint conditioned trajectory prediction. In European conference on computer vision (pp. 759-776). Springer, Cham.
>
> [4] Hu, Y., Chen, S., Zhang, Y. and Gu, X., 2020. Collaborative motion prediction via neural motion message passing. In Proceedings of the IEEE/CVF conference on computer vision and pattern recognition (pp. 6319-6328).
>
> [5] Wang, Chuhua, et al. "Stepwise goal-driven networks for trajectory prediction." IEEE Robotics and Automation Letters 7.2 (2022): 2716-2
>
> [6] Kipf, Thomas, et al. "Neural relational inference for interacting systems." International Conference on Machine Learning. PMLR, 2018.
>
> [7] Makansi, Osama, et al. "You mostly walk alone: Analyzing feature attribution in trajectory prediction." arXiv preprint arXiv:2110.05304 (2021).
>
> [8] Salzmann, Tim, et al. "Trajectron++: Dynamically-feasible trajectory forecasting with heterogeneous data." European Conference on Computer Vision. Springer, Cham, 2020.
>
> [9] Webb, Ezra, et al. "Factorised neural relational inference for multi-interaction systems." arXiv preprint arXiv:1905.08721 (2019).

---

### Author Response · Authors · 2022-08-02
**Response to all reviewers**

We thank the reviewers for their thoughtful and constructive comments. We were encouraged that the reviewers agreed that multiplex attention with progressive layer training is a well-motivated and effective approach for modeling multiagent interactions. In response to excellent feedback, we provide an updated manuscript with a list of improvements and revisions made since the initial submission:

Here we list some improvements and revisions made since the initial submission:
- Inclusion of factorized NRI baseline (Tables 1, 2, 4)
- Inclusion of the results of a newly conducted human experiment on the realism of Figure 5 conditional rollouts (Appendix Section C)
- Table with a summary of relative zero-shot generalization performance (Appendix Table 5)
- Qualitative analysis of the second latent graph layer (Appendix Figure 10, 11)

---

### Meta-Review · Area_Chair_QmgU · 2022-08-21

**Recommendation:** Accept
**Confidence:** Certain

**Metareview:**

The reviewers agreed this paper was presented well and a valuable contribution. We urge the authors to take the reviewers' comments into account in the final version.

Also, please increase the size of the tables -- the font size is quite small (maybe too small).

**Award:**

No

---

### Decision · Program_Chairs · 2022-09-14

Accept